# Decoupling of $\Delta O_2/Ar$ and particulate organic carbon dynamics in near shore surface ocean waters

Sarah Z. Rosengard[1], Robert W. Izett[1], William J. Burt[2], Nina Schuback[3], and Philippe D. Tortell[1,4]

1. Department of Earth, Ocean and Atmospheric Sciences, University of British Columbia, Vancouver, V6T 1Z4, Canada
2. College of Fisheries and Ocean Sciences, University of Alaska Fairbanks, Fairbanks, 99775, USA
3. Institute of Geological Sciences and Oeschger Center for Climate Change Research, University of Bern, Bern, Switzerland
4. Department of Botany, University of British Columbia, Vancouver, V6T 1Z4, Canada

*Correspondence to: Sarah Z. Rosengard (srosengard@eoas.ubc.ca)*

**Abstract.** We report results from two Lagrangian drifter surveys off the Oregon coast, using continuous ship-board sensors to estimate mixed layer gross primary productivity (GPP), community respiration (CR), and net community production (NCP) from variations in biological oxygen saturation ($\Delta O_2/Ar$) and optically-derived particulate organic carbon (POC). At the first drifter survey, conducted in a nearshore upwelling zone during the development of a microplankton bloom, net changes in $\Delta O_2/Ar$ and [POC] were largely decoupled. Significant differences in GPP and NCP derived from $\Delta O_2/Ar$ ($NCP_{O2/Ar}$) and POC ($NCP_{POC}$) time series suggest the presence of large POC losses from the mixed layer. At this site, we utilized the discrepancy between $NCP_{O2/Ar}$ and $NCP_{POC}$ and additional constraints derived from surface water excess nitrous oxide ($N_2O$) to estimate particle export and vertical mixing fluxes, respectively. At the second drifter survey, conducted in lower productivity, density-stratified offshore waters, we also observed significant discrepancies between $\Delta O_2/Ar$ and POC-derived GPP and CR rates. However, net [POC] changes were positively correlated with $\Delta O_2/Ar$ changes, yielding closer agreement in NCP estimates derived from these measurements. This suggests a tighter relationship between production and community respiration, and lower export rates. These results provide insight into the possibilities and limitations of estimating productivity from continuous underway POC and $\Delta O_2/Ar$ data in contrasting oceanic waters. Our observations support the use of diel POC measurements to estimate NCP in lower productivity waters with limited vertical carbon export, and the potential utility of coupled $O_2$ and optical measurements to estimate the fate of POC in high productivity regions with significant POC export.

## 1 Introduction

Marine primary productivity provides the major source of organic carbon to the ocean, supporting the vast majority of marine ecosystem biomass. On short time scales, a large fraction of this fixed organic carbon is converted back to $CO_2$ through community respiration (CR). The difference between gross primary productivity (GPP) and CR – net community production (NCP) – sets an upper limit on the quantity of particulate organic carbon that can be exported out of the mixed layer as sinking particles, transferred to the dissolved organic carbon (DOC) pool, or consumed by upper trophic levels. Accurate assessment of NCP is thus critical to

understanding trophic balance and the fate of organic carbon in the surface ocean. Because
traditional incubation-based approaches to quantify GPP, net primary productivity (NPP) and CR
are labor-intensive and error prone (Gieskes et al., 1979; Fogg and Calvario-Martinez, 1989;
Marra, 2009; Quay et al., 2010), NCP remains challenging to quantify on ecologically-relevant
time and space scales.
In recent years, automated *in situ* measurements of seawater optical properties have been
increasingly used to estimate gross and net primary productivity from changes in optically-
derived surface water POC concentrations (e.g., Graff et al., 2016; Burt et al., 2018). This
approach is based on the relationship between POC concentrations and the particulate fraction of
the beam attenuation coefficient ($c_P$) (Siegel et al., 1989; Stramska and Dickey, 1992; Gardner et
al., 1993; Claustre et al., 1999; Gernez et al., 2011), which can be used to resolve diurnal
variations in POC. This diurnal variability results from the daytime accumulation of
photosynthetically-produced organic carbon, and nighttime loss of fixed carbon through
community respiration, and can thus be used to infer NCP on daily time-scales. The accuracy of
this approach depends on the key assumption that variations in $c_P$ capture most of the variability
in POC concentration, and it has been shown that beam attenuation is most sensitive to particles
with a diameter range of 0.5–20 µm (Stramski and Kiefer 1991; Marra, 2002; Claustre et al.,
2008). To date, most efforts to calculate daily NCP from $c_P$ variability have focused on low
productivity offshore regions, where particle sizes are small and POC losses like particle export
are limited (Claustre et al., 2008; White et al., 2017). These studies have reported good
agreement between optically-derived GPP estimates and independent estimates of NPP from [14]C
incubations (White et al., 2017), suggesting a tight coupling between primary productivity and
mixed layer POC dynamics over daily time scales.
Another approach to NCP quantification is based on autonomous measurements of
surface water dissolved oxygen to argon ratios ($O_2/Ar$). Argon normalization is used to correct
for any physically-induced changes in $O_2$ saturation, such that the derived saturation anomaly,
$\Delta O_2/Ar$, is a tracer of net biological $O_2$ production (Kaiser et al., 2005; Tortell, 2005; Cassar et
al., 2009). At steady-state, and in the absence of significant lateral advection and vertical mixing,
the sea-air flux of excess biologically-produced $O_2$ is equivalent to NCP. With the development
of automated ship-board mass spectrometers, there has been a significant increase in surface
water $O_2/Ar$ measurements, and these have been used to examine $O_2$ variability resulting from
diurnal variations of photosynthesis and respiration, and to infer NCP in a variety of oceanic
ecosystems (Reuer et al., 2007; Stanley et al., 2010; Tortell et al., 2011, 2014; Hamme et al.,
2012; Nicholson et al., 2015; Manning et al., 2017). Recent efforts have shown that NCP
estimates from $\Delta O_2/Ar$ measurements can be corrected for vertical mixing using water column
$N_2O$ measurements as a tracer (Cassar et al. 2014; Izett et al. 2018), but application of this
methodology must assume that lateral advection is negligible.
Combined measurement of mixed layer POC and $O_2$ dynamics holds the potential to
better constrain surface water carbon budgets in biogeochemically dynamic regions, like
upwelling zones, at high spatial and temporal resolution. In net autotrophic systems, an increase
in $\Delta O_2/Ar$ reflects the accumulation of excess photosynthetic $O_2$ in the mixed layer, but provides
no direct insight into the fate of the resulting organic carbon. In the absence of particle export,
grazing or DOC production, an increase in $\Delta O_2/Ar$, corrected for air-sea exchange and vertical
mixing, should be matched by a parallel increase in POC accumulation measured by optical
sensors. By comparison, high POC export, DOC production or grazing coupled to vertical
migrations would act to decouple $\Delta O_2/Ar$ from optically-derived POC measurements in the
mixed layer.
In previous studies, authors have used simultaneous $O_2$ and $c_P$ measurements on
moorings to describe mixed layer $O_2$ and POC dynamics in various marine environments
(Stramska and Dickey, 1992; Kinkade et al., 1999; Dickey and Chang, 2002). However, few
studies to date have compared estimates of primary productivity from simultaneous
measurements on daily time scales. Briggs et al. (2018) and Alkire et al. (2012) were the first to
explicitly combine concurrent measurements of $O_2$ and POC from *in situ* autonomous sensors to
quantify mixed layer productivity during a ~2-month Lagrangian study of the 2008 North
Atlantic spring bloom. Tracking daily changes in mixed layer $O_2$ and POC concentrations, Alkire
et al. (2012) constructed a detailed budget of surface ocean organic carbon throughout the course
of the bloom, using the difference between $O_2$-based NCP and net POC accumulation to assess
the partitioning of NCP into different carbon pools (sinking particles, phytoplankton biomass,
and DOC). Building on this work, Briggs et al. (2018) examined the role of respiration, particle
export, and DOC production in decoupling $O_2$ and POC dynamics through different bloom
stages, demonstrating significant differences between GPP estimates derived from $O_2$, beam
attenuation, and backscatter measurements. To our knowledge, such a detailed examination of $O_2$
and POC dynamics has not been reported for other marine systems.

Here, we present new results from a field study of diel variability in $\Delta O_2/Ar$ and optical

properties in two contrasting near-shore regions of the Subarctic North Pacific. Using ship-board
automated sensors deployed along a Lagrangian drifter track, we resolved fine-scale temporal
patterns in biological oxygen production and POC concentration in a high productivity coastal
upwelling zone over the continental slope and in lower productivity stratified waters offshore.
The biogeochemical differences between both sites provided a unique opportunity to compare
GPP, CR and NCP estimates derived from $\Delta O_2/Ar$ and POC in contrasting trophic regimes. We
expected to observe significant differences between $\Delta O_2/Ar$ and POC-derived GPP, CR, and
NCP estimates in the higher productivity site, reflecting greater carbon export capacity and DOC
production. By comparison, we hypothesized that discrepancies in these rates would be smaller
at the lower productivity site, reflecting a tighter coupling between $O_2$ and POC dynamics.

The results of this investigation have extended findings from the 2008 North

Atlantic bloom to a high productivity coastal upwelling environment, expanding comparisons of
GPP, CR and NCP derived from daily $\Delta O_2/Ar$ and POC variations to a region where vertical
mixing fluxes significantly influence the surface water mass balance. These dynamic systems
play a disproportionately important role in marine biogeochemical cycling, but they pose
significant challenges for interpreting time series of ecosystem metabolism. Furthermore, our
study results further expand applications of a recent field approach to correcting NCP for vertical
mixing (Izett et al., 2018), suggesting that this approach has significant merit in reconstructing
productivity estimates from a variety of mixed layer tracers. We discuss the implications of our
coupled $O_2$-POC measurements for understanding biological carbon cycling in coastal marine
waters, and suggest additional approaches to further improve the utility of coupled $\Delta O_2/Ar$ and
optically-derived organic carbon measurements for evaluating the fate of marine primary
productivity across marine trophic gradients.

**2 Methods**

**2.1 Field site and Lagrangian surveys**

Field studies were conducted on board the R/V *Oceanus* in August 2017, during a
transect through the Northeast Subarctic Pacific Ocean. Two Lagrangian drifters were deployed
off the Oregon coast, allowing us to track diurnal patterns in phytoplankton productivity and
particulate organic carbon cycling in two distinct water masses (Fig. 1). Underway temperature
and salinity measurements, collected by a Seabird SBE 45 thermosalinograph, as well as satellite
(Aqua MODIS) and ship-based chlorophyll-a (Chl-a) observations, were used to guide the
specific location and timing of the drifter deployments. Drifter 1 was deployed on 20 August
2017 (~9:30 PDT), ~40 km from the Oregon coast (44.54° N, 124.58° W), in the vicinity of an
upwelling feature detected based on low sea surface temperature, and elevated salinity and [Chl-
a]. The drifter, consisting of a beacon, GPS transmitter and 5 m drogue, was recovered at ~18:30
on 23 August 2017 (44.40° N, 124.55° W) for a total deployment of 3 days and 9 hours. Upon
recovery, the drogue was missing, implying the potential for some erratic sub-surface drifting
(discussed below). Drifter 2 was deployed approximately 200 km from shore (43.75° N, 126.50
°W) in a relatively warm and low salinity water mass, with low Chl-a concentrations. This
second drifter was deployed at ~07:45 on 24 August 2017, and was recovered after 2 days and
six hours at ~14:00 on 26 August 2017 at 43.80° N, 126.99° W. Because the *Oceanus* lacks a
dynamic positioning system, the ship was not always able to perfectly track the drifter locations.
To correct for these positional offsets, we discarded any observations obtained when the ship
was more than 1.5 km away from the drifter location. This filtered dataset resulted in
measurements every ~15 minutes during the two drifter deployments, yielding 325 and 218
quality-controlled underway observations for drifters 1 and 2, respectively.

**2.2 Underway measurements**

Continuous underway measurements of surface seawater optical properties were
collected using Seabird (formerly Wetlabs) ECO-BB3 and ac-s sensors, following the methods
outlined in detail by Burt et al. (2018). Water was collected from the ship's seawater supply
system with a nominal intake of 5 m depth. Our instrument package included fully automated
data collection, and hourly filtered blanks (0.2μm), which provided measurements of dissolved
seawater optical properties used to infer particulate absorption ($a_p$) and beam attenuation ($c_p$) at
82 wavelengths between 400 and ~735 nm, and backscatter ($b_{bp}$) at 470 nm, 532 nm, and 650

nm. The BB-3 and ac-s measurements were binned into 1-minute intervals. Prior to binning, the absorption and beam attenuation data were first sub-sampled every 50 data acquisition cycles (~12.5 seconds) to enable faster processing time. The optical measurements were accompanied by continuous surface photosynthetically active radiation (PAR) and windspeed data obtained from a Biospherical QSR-220 PAR sensor and Gill WindObserver II ultrasonic wind sensor mounted on the ship's bow.

Chlorophyll-a (Chl-a) concentrations were derived from the particulate absorption line height at 676 nm ($a_{LH}$) (Roesler and Barnard, 2013). Five-minute match-ups between underway $a_{LH}$ and discrete filtered [Chl-a] measurements from the entire cruise transect (Sect. 2.4) were used to derive a best fit coefficient for the linear relationship between $a_{LH}$ and [Chl-a] ($r_2$=0.87, n= 58, p<0.01). Particulate organic carbon (POC) concentrations (µg/L) were derived from particulate beam attenuation at 660 nm ($c_{p,660}$), using the empirical model in Graff et al. (2015). Similarly, phytoplankton organic carbon ($C_{ph}$) concentrations were calculated, using an empirical relationship between particulate backscatter at 470 nm ($b_{bp,470}$) and [$C_{ph}$] in µg/L (Graff et al., 2015). We used a limited set of 5m discrete measurements (n=6) to evaluate the relationship between POC concentrations and $c_p$ at 660nm, and the applicability of the Graff et al. (2015) model to our observations. As shown in Fig. S1, the POC measurements were significantly correlated to $c_p$ ($r_2$=0.88, p<0.05), with a slope and intercept of 391.6 ± 201.6 and 36.7 ± 79.1, respectively. This slope was not significantly different from that of the Graff et al. algorithm (419.8) although our y-intercept was higher. Notwithstanding the relatively small number of discrete POC samples, and some scatter around the regression line, the similarity of our POC-$c_p$ calibration to that reported by Graff et al. (2015) suggests that our optically-derived POC estimates are reasonably robust.

To obtain information on the particle size spectrum, we derived the wavelength-dependent slope of particulate backscatter by fitting the three $b_{bp}$ coefficients (470 nm, 532 nm, 650 nm) to an exponential equation (Stramska et al., 2003; Loisel et al., 2006; Kostadinov et al., 2009). Finally, to assess interference of inorganic minerals on POC, and $C_{ph}$ variability, we calculated the wavelength-specific bulk refractive index ($\eta_p$) from backscatter/total scatter ratios ($\frac{b_{bp}}{c_p - a_p}$) and the wavelength-dependent $c_p$ slope, following the approach of Boss et al. (2001), Twardowski et al. (2001) and Sullivan et al. (2005).

In addition to optical measurements, the seawater biological oxygen saturation anomaly
($\Delta O_2/Ar$) was measured at ~20 second resolution using a membrane inlet mass spectrometer
connected to the ship's seawater intake. The seawater ratio of dissolved $O_2$ and Ar was
determined by diverting a continuous flow of water across a dimethylsilicone membrane
interfaced with a Hiden Analytical HAL20 triple filter quadropole mass spectrometer. The $O_2/Ar$
ratio of air-equilibrated standards ($[O_2/Ar]_{eq}$), incubated at ambient sea surface temperature, was
measured every two hours. Values of $\Delta O_2/Ar$ were thus calculated as the percent deviation of
seawater $O_2/Ar$ measurements from the air-equilibrated ratio, using $\Delta O_2/Ar = 100\%$ *
$([O2/Ar]_{meas} / [O2/Ar]_{eq} - 1)$ (Tortell, 2005; Tortell et al., 2011).

**2.3 Mixed layer depth**

Over the course of both drifter deployments, we conducted regular hydrographic casts
(every six to ten hours) to examine depth profiles of seawater hydrography and biogeochemical
variables. Temperature, salinity, dissolved $O_2$ concentrations and Chl-a fluorescence profile data
from the CTD casts were measured by a Seabird-SBE 38 temperature sensor, Seabird-SBE 4
conductivity sensor, SBE 43 dissolved $O_2$ sensor, and a Seabird ECO fluorometer, respectively,
and binned into 1 m intervals. Vertical profiles at the drifter 1 site showed relatively weak
density stratification, likely as a result of recent upwelling. For this reason, we estimated mixed
layer depths ($z_{mld}$) based on visible inflection points in the dissolved $[O_2]$, fluorescence and
density profiles, assuming that dissolved $O_2$ concentrations and fluorescence are relatively
uniform in the mixed layer. Within a single CTD cast, mixed layer depths varied by up to 28%
across all three profile measurements. The [Chl-a] fluorescence profiles had the most well-
defined inflection points, and we thus used these data to estimate $z_{mld}$ at all casts. Excluding
fluorescence profiles from the first day (Sect. 3.1), and two casts at 6am and midnight on second
and third 24-hour intervals, respectively, which displayed relatively noisy density profiles, an
average $z_{mld}$ value ($19 \pm 2$ m) was derived and applied to all subsequent analyses.
In comparison to the drifter 1 site, CTD cast profiles during drifter deployment 2 showed
larger density gradients. We thus computed $z_{mld}$ using a density difference criterion of 0.25
$kg/m_3$ (Thomson et al., 2003; de Boyer Montégut et al., 2004) from median values within the
upper-most 4–6 m of the profile. We found that this critical density criterion was necessary to
capture the depth of inflection in $O_2$ and [Chl-a]. In all CTD casts except one, density difference-
based $z_{mld}$ values were within 5 meters of the values derived from the inflection points on density
profiles. An average $z_{mld}$ value estimated from the density-difference approach ($22 \pm 5$ m) was
applied to all subsequent analyses.

**2.4 Discrete samples**

Concentrations of phosphate ([$PO_4^{3-}$]), dissolved silica ([$SiO_2$]), and nitrate and nitrite

([$NO_3^- + NO_2^-$], were measured in seawater samples collected from daily Niskin bottle casts.
Following collection, nutrient samples were filtered through 0.2 µm pore polycarbonate
membranes and immediately frozen at -80°C on board the ship. These samples were stored at -
20°C until subsequent colorimetric laboratory analyses (Murphy and Riley, 1962; Riley, 1977)
with a Lachat QuikChem 8500 Series 2 Flow Injection Analysis System.

Concentrations of dissolved oxygen ($O_2$) and nitrous oxide ($N_2O$) were measured in

discrete samples collected in Niskin bottles during both drifter deployments (Fig. S2), following
methods outlined in (Capelle et al., 2015). These $N_2O$ measurements were used to correct NCP
estimates for vertical mixing (see Sect. 2.6), following the approach described by Cassar et al.
(2014) and Izett et al. (2018). Profile samples from the first day of drifter deployment 1 (August
20) were omitted from calculations, as underway surface temperature and salinity measurements
indicated intrusion of an external water mass (further discussed in Sect. 3.1) (Fig. S3). Three
profiles collected from 12:00 (PDT) CTD casts during the following three days of the
deployment (August 21, 22 and 23) were applied to the NCP mixing correction at drifter station
1 (Sect. 2.6.1).

Surface (~5 m) discrete seawater samples were collected either from Niskin bottles or

from the ship's surface seawater intake system for HPLC analysis of Chl-a concentrations and
other phytoplankton pigments. Single or duplicate samples were filtered onto 25 mm GF/F
filters, flash-frozen in liquid nitrogen, and stored at -80°C until analysis, following the
methodology described in Schuback et al. (2016). Additional samples were collected from the
seawater intake for size-fractionated Chl-a analysis (Zeng et al., 2018). These samples were
filtered through stacked 47 mm filters (0.2 µm, 2 µm and 20 µm pore size) separated by a mesh
spacer. Filtered samples were extracted in 5 mL of 90% acetone at 4°C until analysis within 24–
48 hours using a Turner Trilogy Fluorometer on board the ship.

Discrete samples for POC analysis were collected at two depths from several CTD casts.

Surface samples were collected at both drifter sites from 5 m depth, while deeper samples were
collected at near the base of the euphotic zone (~1% PAR), corresponding to 40–60 m at drifter
site 1, and 100–120 m at drifter site 2. POC samples (~1–4 L) were filtered through a pre-
combusted (450 °C) Whatman GF/F filter (nominal pore size ~ 0.7 μm), and stored at -80°C
until laboratory analysis. Prior to analysis, samples were thawed and dried at 50°C overnight,
fumigated with concentrated hydrochloric acid for 48 hours, and dried again at 50°C overnight.
POC concentrations in samples (and blank combusted filtered treated as described above) were
quantified using an *Elementar* vario MICRO cube CHNS analyzer. Blank-corrected discrete
POC concentrations were used to validate application of the [POC] model in Graff et al. (2015)
to our underway $c_p$ data (Sect. 2.2; Fig. S1).

**2.5 Net Primary Productivity**

Daily-integrated net primary productivity (NPP) was calculated in two ways. First,

carbon uptake was determined from 24-hour $_{14}$C-incubations with 5 m triplicate seawater
samples collected from early morning CTD casts. Measurements were made on two different
mornings during drifter deployment 1 and on one morning during drifter deployment 2. The
measurements were conducted following the protocol outlined in Hoppe et al. (2017). Depth-
integrated NPP was calculated by multiplying the derived 24-hour volumetric carbon fixation
rate by the average mixed layer depth for the respective drifter period.

Second, daily-integrated net primary productivity was also estimated as a product of $[C_{ph}]$

values derived from $b_{bp}$, and phytoplankton growth rates according to the carbon-based
productivity model (CbPM) (Behrenfeld et al., 2005; Westberry et al., 2008; Graff et al., 2016;
Burt et al., 2018). In these calculations, daily-averaged $[C_{ph}]$, $[Chl-a]/[C_{ph}]$, and mixed layer
irradiance ($E_g$) calculated from the MODIS-derived surface PAR matched to drifter location
were used to calculate growth rates and NPP every 24 hours. Chlorophyll-a concentrations were
derived from absorption line height (Sect. 2.2), $[C_{ph}]$ values from $b_{bp}$, and light extinction
coefficients ($K_d$) obtained from [Chl-a] to calculate $E_g$ (Morel et al., 2007). An average mixed
layer depth for each drifter period was applied to estimate mixed layer NPP (Sect. 2.3).

**2.6 Quantification of GPP, CR and NCP**

Gross primary productivity (GPP), community respiration (CR) and net community

production (NCP) rates were calculated based on linear regressions of $\Delta O_2/Ar$ and POC against
time over subsequent day (D) and night (N) intervals during both drifter deployments. Daytime
was defined as the period during which PAR levels exceeded 20 µmol quanta $m^{-2}s^{-1}$. The average
length of the day-time period was $13.6 \pm 0.14$ hours over the two drifter deployments. In the
following sections, $t_d$ represents the day length normalized to 24 hours, and $t_n$ analogously
represents the fractional night length, equivalent to $1-t_d$. All daily rates were integrated through
the mixed layer using the average $z_{mld}$ for each drifter period, as described in Sect. 2.3.

**2.6.1 $O_2/Ar$-derived rates**

Quantification of $GPP_{O2/Ar}$, $CR_{O2/Ar}$, and $NCP_{O2/Ar}$ rates from diurnal cycles in $\Delta O_2/Ar$

(Ferrón et al., 2015) requires corrections for gas exchange and, potentially, vertical mixing
fluxes. For these calculations, we first computed the rate of change in $\Delta O_2/Ar$ ($dO_{2Bio}/dt$) using
linear regression analysis within successive daytime or nighttime intervals. We then derived
estimates for the air-sea gas exchange ($J_{ex}$) and vertical mixing fluxes ($F_{mix}$) over the respective
day or night interval to isolate the NCP contribution to observed $\Delta O_2/Ar$ changes (Izett et al.,
2018; Tortell et al., 2014). Net $O_2$ production rates were converted into carbon units using a
photosynthetic quotient (PQ) for new production of 1.4 for drifter period 1 calculations and a PQ
for regenerated production of 1.1 for drifter period 2 (Laws, 1991). We assumed that CR rates
were constant over each respective day length period (i.e. $t_d + t_N$).

$$NCP_{\frac{O2}{Ar},D\ or\ N} = z_{mld} \left. \frac{dO_{2bio}}{dt} \right|_{D\ or\ N} + J_{ex}|_{D\ or\ N} + F_{mix} \hspace{2cm} (1)$$

$$GPP_{O2/Ar} = \frac{t_d(NCP_{\frac{O2}{Ar},D} - NCP_{\frac{O2}{Ar},N})}{PQ} \hspace{2cm} (2a)$$
$$CR_{O2/Ar} = \frac{NCP_{\frac{O2}{Ar},N}}{PQ} \qquad (2b)$$
$$NCP_{\frac{O2}{Ar},24hr} = \frac{t_d NCP_{\frac{O2}{Ar},D} + t_N NCP_{\frac{O2}{Ar},N}}{PQ(t_d+t_N)} \qquad (2c)$$

$$O_{2bio} = \Delta\frac{O_2}{Ar}\frac{1}{100\%}O_{2eq} \qquad (3)$$

$$J_{ex} = k_{o2}O_{2bio} \qquad (4)$$

$$F_{mix,O2/Ar} = k_{mix}\frac{dO_{2bio}}{dz} = k_{N2O}N_2O_{bio}\frac{dO_{2bio}}{dN2O_{bio}} \qquad (5)$$

$$k_{mix} = k_{N2O}N_2O_{bio}(\frac{dN2O_{bio}}{dz})^{-1} \qquad (6)$$

$$N_2O_{bio} = N_2O_{meas} - N_2O_{eq} - N_2O_{thermal} \qquad (7)$$

Equilibrium concentrations of $O_2$ and $N_2O$ ([$O_2$]$_{eq}$ and [$N_2O$]$_{eq}$) were calculated using the

salinity and temperature-dependent equations of Garcia and Gordon (1992) and Weiss and Price
(1980), respectively, and sea surface temperature and salinity from the ship's thermosalinograph.
Estimates of surface excess $N_2O$ saturation, [$N_2O$]$_{bio}$, included a heat flux correction to account
for solubility changes (Keeling and Shertz, 1992; Jin et al., 2007; Izett et al., 2018). Non-
weighted piston velocities ($k_{O2}$ and $k_{N2O}$) were calculated using the diffusive air sea gas flux and
Schmidt number parameterizations of Wanninkhof (2014) and Raymond et al. (2012), and ship-
based wind speed data 10 m above the sea surface. Daytime and nighttime estimates for the gas
exchange term, $J_{ex}$, were calculated using day/night average [$O_2$]$_{eq}$, $\Delta O_2/Ar$, and $k_{O2}$ values.
Vertical gas gradients ($\frac{dN2O_{bio}}{dz}$ and $\frac{dO_{2bio}}{dN2O_{bio}}$) were estimated from our measurements over the
upper 100 m of the water column, following Izett et al. (2018).

At drifter site 1, daily $F_{mix}$ values were calculated using daily [$N_2O$]$_{bio}$, daily vertical

gradient and daily average $k_{N2O}$ values (Sect. 2.4). Denitrification should not have been a source
of $N_2O$ within the upper 100 m of the water column because measured $O_2$ concentrations were
consistently greater than the threshold value of ~50 mmol m$_{-3}$ (e.g., Hopkinson and Barbeau,
2007). Likewise, we assumed no lateral advection of $N_2O$ into drifter site 1, as there were little
differences in the mixing ratio $[O_2]_{bio}/[N_2O]_{bio}$ across profile measurements (Fig. S2). While the
August 22 CTD cast did exhibit a more anomalous $[O_2]_{bio}/[N_2O]_{bio}$ profile relative to the other
two cast profiles, inclusion of these data had little impact on the vertical mixing correction. At
drifter site 2, we assumed that vertical mixing was negligible due to the presence of strong
density stratification, and therefore did not calculate a mixing flux correction at this site. In any
case, the presence of a sub-surface $O_2$ maximum (Fig. S2) would significantly limit the
application of the $N_2O$ correction (Izett et al., 2018).

**2.6.2 Optically-derived rates**

We used the approach of Claustre et al. (2008) and White et al. (2017) to calculate daily-
integrated $GPP_{POC}$, $CR_{POC}$, and $NCP_{POC}$ from daytime and nighttime changes in POC (dPOC/dt),
derived from linear regressions of POC concentrations against time through day and night
intervals. In certain ocean environments, $NCP_{POC}$ will not equate to $NCP_{O2/Ar}$ as a result of
additional POC sinks, including export, grazing and DOC production. Under these conditions,
$CR_{POC}$ includes these loss term, and therefore $NCP_{POC}$ more accurately reflects net POC
accumulation. Nonetheless, for consistency with previous studies, we use the term $NCP_{POC}$ to
describe the quantities computed in Eq. 8.

$$NCP_{POC,D\ or\ N} = z_{mld}\ \frac{dPOC}{dt}\bigg|_{D\ or\ N} + F_{mix(POC)} \qquad (8)$$

$$GPP_{POC} = \frac{t_d(NCP_{POC,D} - NCP_{POC,N})}{PQ} \qquad (9a)$$
$$CR_{POC} = \frac{NCP_{POC,N}}{PQ} \qquad (9b)$$
$$NCP_{POC,24hr} = \frac{t_d NCP_{POC,D} + t_N NCP_{POC,N}}{(t_d + t_N)} \qquad (9c)$$

The presence of significant upwelling at drifter site 1 provides additional complexity in the
estimate of NCP from optically-derived POC measurements.  In particular, vertical transport of
particle-poor seawater from below the mixed layer into the surface could dilute the $c_p$ signal used
to derive POC concentrations (Stramska and Dickey, 1994). To address this, we applied the
vertical mixing term, $k_{mix}$, derived from Eq. (6) to estimate the average daily dilution effect on
mixed layer POC concentrations through drifter period 1:

$$F_{mix,POC} = k_{mix} \frac{dPOC}{dz} \tag{10}$$

The term d[POC]/dz represents the vertical gradient in [POC], derived from daily average POC
concentrations measured in Rosette samples at 5 m and near the base of the euphotic zone, below
the mixed layer (40–60 m) (Sect. 2.4). The dz term was calculated as the difference between the
average mixed layer depth from all CTD casts and the daily average shallowest depth of
minimum particle concentrations based on beam transmission profiles. The uncertainty
associated with this gradient calculation is addressed in the discussion section. At drifter site 2,
$F_{mix,POC}$ was considered negligible (Sect. 2.6.1) due to the high density stratification of the water
column.

In total, three sets of 24-hour GPP, CR and NCP values were calculated during the drifter

1 deployment from the three pairs of consecutive day and night intervals, starting with the first
night interval and ending with the last day interval. We excluded the first day-time interval from
our calculations, due to the erratic salinity values observed during the first day of this drifter
deployment (Sect. 3.1; Fig. S3). Because the drifter period was terminated prior to sunset, the
last day interval was 1.6 hours shorter than the average daytime duration. For the second drifter
deployment, two sets of GPP, R and NCP values were calculated from consecutive day and night
intervals, starting with the first daytime interval and ending with the last nighttime interval. The
initiation of the drifter period occurred after sunrise, so the first day interval was 1.1 hours
shorter than the average daytime duration.

**2.6.3 Integration time scales**

The approach to calculating NCP on the basis of linear regressions utilizes the high

temporal resolution of our data set. We compared our results from Sects. 2.6.1 and 2.6.2 to NCP
values calculated using several of other integration time scales. Following studies that have
calculated daily NCP values from "instantaneous" rates of change (e.g., hourly rates in Hamme
et al., 2012; Tortell et al., 2014), we divided our NCP calculations into shorter increments. Given
that the average measurement interval was ~15 minutes (after removing values where the ship
was not sufficiently close to the drifter; Sect. 2.1), we calculated NCP within three-hour
intervals:

$$NCP_{\frac{O2}{Ar},3hr} = \frac{3}{24}\left[z_{mld}\left(\frac{dO_{2bio}}{dt}\right)_{3hr} + J_{ex,3hr}\right] \quad \text{(11a)}$$
$$NCP_{POC,3hr} = z_{mld}\left[\frac{3}{24}\left(\frac{dPOC}{dt}\right)_{3hr}\right] \quad \text{(11b)}$$

For each day of the drifter periods, eight consecutive three-hour NCP values were summed into a
24-hour period to yield daily NCP estimates. We then applied the vertical mixing correction to
these daily estimates (refer to Eqs. 5, 6, 10), since the correction was only available on a daily
basis given the lower sampling resolution of [$N_2O$] and [POC] profiles. We also calculated daily
NCP using the difference between $\Delta O_2/Ar$ or [POC] between two time points at the beginning
and end of each 24-hour period (similar to approach in Alkire et al. 2012; and Barnes and
Antoine, 2104). Finally, we calculated a single daily NCP rate per drifter period using the linear
regression of $\Delta O_2/Ar$ and [POC] against time over the entire drifter period. For these latter two
approaches, the 24-hour average and drifter-period average of relevant terms in Eqs. 1-9 were
used to calculate NCP, respectively.

**2.7 Error analysis**

Errors for all estimates of net primary productivity (CbPM-NPP, $_{14}C$-NPP) and net

community production (NCP$_{O2/Ar}$, NCP$_{POC}$) were propagated from uncertainties associated with
all variables used for the computations. Error estimates for time-averaged variables were
generally represented by the standard deviation, as we assumed that this significantly exceeded
the error of the individual variables prior to averaging. The uncertainty in $z_{mld}$, derived from the
standard deviation of mixed layer depths across individual CTD casts, was 2 m for drifter site 1
and 5 m for drifter site 2 (Sect. 2.3). Small uncertainties in $t_D$ and $t_N$ were calculated as the
standard deviations of all day or night lengths measured during both drifter periods (0.14 and
0.10 hours, respectively). Mean relative errors of [Chl-a] and [$C_{ph}$] from Burt et al. (2018), and
mean relative standard deviations in MODIS-derived daily surface PAR values were propagated
to calculate the error in CbPM-NPP. The standard deviations of triplicate 24-hour 14C uptake
incubations were propagated to calculate the error in 14C-NPP estimates. The uncertainties in
14C-NPP values are likely underestimated, as they do not account for bottle effects, as discussed
in Sect. 4.3.

For calculating error in NCP, uncertainties in $dO_{2bio}/dt$ and dPOC/dt were derived from

the confidence interval of the best-fit slope of linear regression of each variable against time.
Standard deviations of averaged $\Delta O_2/Ar$, $k_{O2}$, and $k_{N2O}$ values, and the mean relative errors of
$[N_2O]_{meas}$, $[N_2O]_{Eq}$, $[N_2O]_{thermal}$, and $\frac{dO_{2bio}}{dN2O_{bio}}$ reported in Izett et al. (2018), were propagated into
the mixing correction errors for $NCP_{O2/Ar}$ and $NCP_{POC}$. The error in $\frac{dN2O_{bio}}{dz}$ was calculated as the
confidence interval of the best fit slope extracted from a linear regression of pooled drifter 1
$[N_2O]_{bio}$ values against depth. Finally, to account for uncertainty in the photosynthetic quotient
(PQ), we applied a PQ variability of 0.1 to $NCP_{O2/Ar}$ calculations, following Laws (1991).

**3 Results**

**3.1 Water mass properties**

Ship-board underway measurements revealed clear differences in hydrographic and

biogeochemical characteristics between the water masses sampled by the two drifters. Surface
water properties at drifter site 1 reflected the presence of a recently upwelled water mass that was
relatively cold (11.8 ± 0.4 °C), saline (32.6 ± 0.04 g/kg), and nutrient-rich (Figs. 1, S3, S4). The
Pacific Fisheries Environmental Laboratory's coastal upwelling index at 45°N, 125°W was
positive throughout drifter period 1. In contrast, the water mass tracked by the second drifter
deployment was warmer (17.5 ± 0.1°C) and fresher (31.8 ± 0.05 g/kg), with lower average mixed
layer nutrient concentrations.

Examination of surface water hydrographic properties during the two drifter deployments

suggest that both drifters tracked a relatively homogenous water mass, excluding a period of
salinity variability during the first day of drifter deployment 1, and several transient temperature
and salinity excursions after the second night of this deployment (grey patches in Fig. S3). These
features indicate potential intrusion of an external water mass, possibly a result of loss of the
drifter drogue (Sect. 2.1). Observations during these periods were thus removed from the data set
prior to analysis to ensure the most accurate calculation of productivity rates. Outside of these
intervals, variability in salinity (drifter 1: 32.5–32.7 g/kg; drifter 2: 31.8–31.9 g/kg) was small
during both drifter deployments. Variability in sea surface temperature was also limited (drifter
1: 11.2–13.0 °C, drifter 2: 17.3–17.7 °C), and largely reflected a diurnal variation of warming
and cooling, which was particularly evident for drifter period 2.

Temporal differences in CTD cast profiles point to some variation in mixed layer depth

($z_{mld}$) during both drifter deployments. In general, there were no multi-day trends or regular
diurnal patterns in $z_{mld}$ through both periods, suggesting that transient shifts in water column
turbulence likely contributed to changes in the shape of temperature, salinity, dissolved oxygen
and fluorescence profiles. Average $z_{mld}$ values, calculated over each drifter period, had relatively
low relative standard deviations (<25%) and were applied to all subsequent calculations (Sect.
2.3). A sensitivity analysis, not shown, indicated that the choice of mixed layer depth using
different criteria (i.e., fluorescence profiles, density profiles and the density difference criterion)
and different time scales of integration (i.e., daytime/nighttime, 24 hour, and multi-day) did not
significantly impact the results discussed below.

Average mixed layer nutrient concentrations fluctuated during both drifter deployments,

but did not exhibit regular diurnal cycles (Fig. S4). At drifter site 1, concentrations ranged from
0.74 to 0.85 µM phosphate, 7.8 to 9.0 µM nitrate and nitrite, and 9.2 to 11.1 µM dissolved silica,
excluding day 1 of the drifter deployment and anomalously high concentrations measured during
a noisy CTD cast at midnight on the last day of the deployment. Excluding these outliers, a
significant ($p < 0.05$) linear regression of each nutrient concentration against time revealed that
phosphate concentrations decreased by ~0.07 µM, [$NO_3^-$ + $NO_2^-$] decreased by 0.9 µM, and
[$SiO_2$] decreased by 1.2 µM over the three-day drifter period, roughly in Redfield ratio
proportions (Sect. 3.4). Nutrient concentrations varied less at site 2, from 0.08–0.10 µM [$PO_4^{3-}$],
0.29–0.61 µM [$NO_3^-$ + $NO_2^-$], and 1.2–1.7 [$SiO_2$]. While [$PO_4^{3-}$] and [$SiO_2$] increased
significantly ($p < 0.05$) by 0.015 µM and 0.48 µM, respectively, these changes were small
compared to the shift observed during drifter period 1, and did not reflect Redfield ratio
proportions. It is possible that intrusions of an external water mass with slightly elevated nutrient
concentrations have contributed to the small increase in [$PO_4^{3-}$] and [$SiO_2$] measured during
these CTD casts, even though we assume that such effects on our derived productivity estimates
are negligible based on inspection of underway temperature and salinity data (Fig. S3).

**3.2 Biogeochemical comparisons between drifter sites**

Elevated nutrient concentrations at the drifter 1 site supported high productivity and the

accumulation of phytoplankton biomass, as indicated by elevated chlorophyll-a ([Chl-a]= 0.66–
1.5 µg/L), phytoplankton carbon ([$C_{ph}$]= 83–115 µg/L) and particulate organic carbon
concentrations ([POC]= 130–261 µg/L) (Figs. 2a–c). We observed [$C_{ph}$]/[Chl-a] ratios ranging
from 68–143 g/g, with a median value of 85 g/g (Fig. 2f). Using the carbon-based production
model (CbPM; Sect. 2.5) and daily-averaged mixed layer PAR derived from satellite values
matched to drifter location (within 5 km), these [$C_{ph}$]/[Chl-a] ratios translate into phytoplankton
growth rates ranging from 0.75–0.94 d-1. At the second drifter site, phytoplankton productivity
and biomass were significantly lower in the nutrient-poor waters ([Chl-a]= 0.06–0.21 µg/L,
[$C_{ph}$]= 11–17 µg/L, and [POC]= 25–38 µg/L). Ratios of [$C_{ph}$] to [Chl-a] at site 2 were
significantly higher (p<0.05) than those observed at site 1, ranging from 69 g/g to 203 g/g, with a
median value of 108 g/g. The higher ratios may reflect reduced cellular [Chl-a] associated with
greater nutrient limitation, higher daily-integrated PAR, and proportionally more picoplankton
than microplankton at drifter site 2 (Westberry et al., 2008; Hirata et al., 2011; Graff et al., 2016;
Burt et al., 2018). Median PAR levels were higher and less variable at site 2, in part contributing
to lower variability in CbPM-based growth rates, which ranged from 0.81 to 0.85 d-1.

Several lines of evidence suggest that the phytoplankton assemblage at drifter site 1 was

enriched in large-celled phytoplankton, as compared to drifter site 2. The wavelength-dependent
slope of particulate backscatter ($b_{bp}$) was lower at site 1 (range: 1.4 to 1.6, median: 1.5) than at
site 2 (range: 1.9–2.3, median: 2.1) (Fig. 2d), suggesting proportionally larger particle sizes
(Stramska et al., 2003; Kostadinov et al., 2009). This observation is supported by size-
fractionated Chl-a measurements. During the drifter 1 deployment, the >20 µm size fraction
(Sect. 2.4), increased from 21 % to 46 % of the total Chl-a pool, indicating the enrichment of
large phytoplankton in the assemblage. Indeed, pigment-based estimates of phytoplankton
taxonomic composition and size class (Hirata et al., 2011; Zeng et al., 2018) suggested that
relative diatom and microplankton abundances exceeded 50% on the final sampling time point.
By comparison, size-fractionated [Chl-a] and HPLC analyses from drifter 2 indicated a lower
proportion of large-celled phytoplankton, with 9–15% of total Chl in the >20 µm size fraction,
and diatoms and micro-plankton comprising 19–29% of the phytoplankton assemblage. The
proportion of picoplankton increased through time at drifter site 2 from 31–50% of total [Chl-a],
alongside slight increase in $b_{bp}$ slope, indicating accumulation of smaller particle sizes (Fig.
S3d). Finally, median bulk refractive index values across three wavelengths (470 nm, 532 nm,
650 nm) were higher at site 1 (1.08–1.11) than at site 2 (1.02–1.04) (Fig. S3e), which is
consistent with a greater proportion of diatom-derived amorphous silica in the particle pool
(Lide, 1997; Twardowski et al., 2001).

**3.3 Diurnal variability and primary production**

As shown in Fig. 3a, clear diurnal cycles in biological oxygen saturation ($\Delta O_2/Ar$) were
observed during both drifter deployments. Generally, values of $\Delta O_2/Ar$ increased from dawn to
dusk and decreased from dusk to dawn, yielding positive slopes of linear regressions of $\Delta O_2/Ar$
against time in the daytime, and negative slopes at night (Fig. S5a). During drifter deployment 1,
this diurnal cycle was superimposed on a longer-term increase in biological $O_2$ saturation as
under-saturated values returned toward atmospheric equilibrium. At least part of this increase is
attributable to gas exchange, which would act to erase $O_2$ under-saturation in the mixed layer
caused by recent upwelling. However, calculation of the sea-air $O_2$ flux shows that, except for
the first 24-hour period, only a small amount of the daily increase in $\Delta O_2/Ar$ can be explained by
gas exchange (absolute value of $J_{ex} < 10$ mmol $O_2$ m$_{-2}$ d$_{-1}$) (Table 1). Thus, the temporal change
in $\Delta O_2/Ar$ can be attributed to a primarily biological source. The magnitude of this increase is
further underestimated because of vertical upwelling of deep oxygen-poor waters, which would
act to dampen the increase in $\Delta O_2/Ar$ through time. After accounting for a mixing correction
ranging between 22 and 97 mmol m$_{-2}$ d$_{-1}$ $O_2$ (equivalent to 16 to 70 mmol m$_{-2}$ d$_{-1}$ C when
assuming a photosynthetic quotient of 1.4), daily-integrated gross primary productivity
($GPP_{O2/Ar}$) ranged from 270 to 358 mmol C m$_{-2}$ d$_{-1}$, and community respiration ($CR_{O2/Ar}$) rates
ranged from 74 to 172 mmol C m$_{-2}$ d$_{-1}$ (Table 1).
Examination of the diel variability in POC and Chl-a during drifter period 1 revealed
significant differences in the behavior of these variables as compared to $\Delta O_2/Ar$ (Fig. 3b, c).
Namely, $\Delta O_2/Ar$ increased during the first drifter deployment, whereas [POC] and [Chl-a] values
decreased. We estimated that vertical mixing ($F_{mix,POC}$), accounted for 12 to 68 mmol m$^{-2}$ d$^{-1}$ C of
these daily changes in [POC], similar to the magnitude of the mixing correction for $\Delta O_2/Ar$
variability (Table 1). After taking mixing into account, daily-integrated GPP$_{POC}$ decreased from
242 mmol m$^{-2}$ d$^{-1}$ on day 1 to 98 mmol m$^{-2}$ d$^{-1}$ on day 3, while CR$_{POC}$ rates ranged from 77 to
147 mmol m$^{-2}$ d$^{-1}$.

Calculated daily averaged net primary productivity (NPP) were lower than GPP$_{O2/Ar}$.

Rates derived from the CbPM model (Sect. 2.5), declined from 147 mmol C m$^{-2}$ d$^{-1}$ on day 1 of
drifter deployment 1 to 112 mmol C m$^{-2}$ d$^{-1}$ on day 3 (Table 1), reflecting the trend in Chl-a
concentrations used to derive NPP (Fig. 3c). The CbPM-derived NPP estimate was similar to that
obtained in $^{14}$C incubations (150 $\pm$ 18 mmol C-m$^{-2}$d$^{-1}$) within the first 24 hours of drifter
deployment 1. However, $^{14}$C-based NPP estimates on the third day of the deployment (49 $\pm$ 8
mmol C-m$^{-2}$d$^{-1}$) were about two-fold lower than those obtained from CbPM calculations.

Dissolved oxygen and POC dynamics at drifter site 2 differed significantly from those

observed at site 1. Compared to the drifter site 1, diel variability in $\Delta O_2/Ar$ and [POC] was more
tightly coupled during the second drifter deployment (Fig. 3a, b). Both $O_2/Ar$ and [POC]
displayed regular diurnal variations, increasing in the daytime to a maximum around dusk and
decreasing at night to a minimum around dawn (Fig. S5a, b). Over the full drifter deployment,
concentrations of Chl-a and, to a lesser extent, POC, decreased, in contrast to $\Delta O_2/Ar$, which
remained relatively constant across days. Daily-integrated GPP$_{O2/Ar}$ values ranged from 108 to
219 mmol C m$^{-2}$ d$^{-1}$ and CR$_{O2/Ar}$ rates ranged from 82 to 186 m$^{-2}$ d$^{-1}$. POC-derived values were
considerably lower and less variable, from 41 to 38 for GPP$_{POC}$ and 36 to 44 for CR$_{POC}$ (Table
1). NPP derived from CbPM calculations was 22 mmol C m$^{-2}$ d$^{-1}$ on the first day of the drifter
period and 18 mmol C m$^{-2}$ d$^{-1}$ on the second day, while NPP calculated from one $^{14}$C bottle
incubation during the first day of the drifter 2 deployment was 12 $\pm$ 4 mmol C m$^{-2}$ d$^{-1}$, showing
good agreement with the CbPM calculations.

**3.4 Net community production**

Daily net community production (NCP) rates were calculated using linear regressions of

$\Delta O_2/Ar$ and POC over day and night intervals, corrected for gas exchange and vertical mixing
(Sect. 2.6.1, 2.6.2). During drifter period 1, $NCP_{O2/Ar}$ and $NCP_{POC}$ exhibited contrasting trends, as
$NCP_{O2/Ar}$ remained >100 mmol C $m^{-2}$ $d^{-1}$ throughout, while $NCP_{POC}$ declined to negative values
on the second and third days (Table 1; Fig. 4). At drifter period 2, we observed closer agreement
between NCP values. $\Delta O_2/Ar$-derived NCP ranged from -12 to 33 mmol C $m^{-2}$ $d^{-1}$ over two
consecutive 24 hour periods, while $NCP_{POC}$ values ranged from -3 to 1 mmol C $m^{-2}$ $d^{-1}$. These
lower rates at drifter site 2 are consistent with the lower observed phytoplankton biomass and
nutrient concentrations.

Additional constraints on NCP during drifter period 1 can be derived from examining

nutrient drawdown. Because vertical upwelling of nutrient-replete waters would dampen the
magnitude of observed nutrient drawdown over time (Sect. 3.1), we used the derived $k_{mix}$ from
Eq. 6 and a best-fit vertical gradient in nutrient concentrations between the mixed layer and 100
m (Sect. 2.4) to account for this mixing flux. This correction increases the cumulative three-day
nutrient drawdown by 2.1 to 2.6 times. Over the three-day drifter deployment (Sect. 3.1), surface
Si, N and P concentrations declined in a ratio of 17: 13: 1, which is consistent with the
stoichiometry expected for organic matter produced by a diatom-rich assemblage (Brzezinski et
al., 1998; Turner et al., 1998; Brzezinski, 2004). Assuming that the observed decrease in $SiO_2$
concentrations over the three days is attributable to growth of diatoms in the mixed layer, and
applying a stoichiometric ratio of 106 C: 16 Si, we estimate an average C fixation rate of ~128
mmol C $m^{-2}$ $d^{-1}$ for the drifter period. This value is consistent with $NCP_{O2/Ar}$ rates, which were
137 mmol C $m^{-2}$ $d^{-1}$ on average over three days, but significantly greater than $NCP_{POC}$ estimates
(7 mmol C $m^{-2}$ $d^{-1}$ on average) (Table 2).

Table 2 summarizes comparisons among NCP values calculated using day/night linear

regressions of $\Delta O_2/Ar$ and POC against time, and other approaches described in Sect. 2.6.3. In
general, our main conclusions were not significantly altered by different calculation methods.
NCP values derived from one linear regression over each drifter period agreed well with the
average of two (drifter 2) to three (drifter 1) daily NCP values calculated via the other
approaches. Small differences between linear regression-based NCP values and both NCP
calculated from either 3-hour increments or two time points are likely due to the effect of lower
signal to noise in $\Delta O_2/Ar$, $[O_2]_{bio}$ and [POC] values utilized in these latter two approaches. Thus,
the following discussion focuses on productivity rates derived from day/night linear regressions
(i.e., Eqs. 1 and 8) because they utilize all data points while minimizing uncertainty in the
derived rates of change. The exception is the $NCP_{O2/Ar}$ value calculated for day 1 of drifter period
2 using the daytime/nighttime linear regression method. By this approach, we calculated
$NCP_{O2/Ar}$ as 26 mmol C m$_{-2}$ d$_{-1}$, even though the time series in Fig. 3a clearly indicates a net
decrease in $\Delta O_2/Ar$ over the 24-hour period, and all other $\Delta O_2/Ar$-based NCP calculations (Sect.
2.6.3) yielded negative values. For the discussion, Table 1 and Fig. 4, the NCP value derived
from the integrated 3-hour increments represents net community production during this
particular interval.

**4 Discussion**

The results from our Lagrangian surveys illustrate diurnal dynamics in two contrasting
productivity regimes off the Oregon coast. Biogeochemical properties during the first drifter
deployment suggested a dynamic, highly productive phytoplankton community, influenced by
upwelling and elevated mixed layer nutrient concentrations (Figs. 1, S4). Several lines of
evidence imply the presence of a developing diatom bloom at this site (Sect. 3.2; Figs. 2, 3).
Increasing mixed layer biological oxygen saturation ($\Delta O_2/Ar$) was contrasted by a general
decrease in particulate organic carbon (POC) concentrations, suggesting a significant decoupling
between $O_2$ and POC dynamics. This was reflected in significant differences between $\Delta O_2/Ar$-
derived gross primary productivity (GPP) and net community production (NCP) rates derived
from $\Delta O_2/Ar$ and POC measurements (Figs. 4, 5; Table 1). In contrast, biogeochemical
properties during the second drifter deployment were indicative of a lower productivity, nutrient-
limited phytoplankton assemblage, with near-zero $\Delta O_2/Ar$ values reflecting a close balance
between water column photosynthesis and respiration (Fig. 3a). Relative to the drifter 1 site,
diurnal variations in $\Delta O_2/Ar$ and POC were more closely coupled, while phytoplankton biomass
($C_{ph}$) and chlorophyll-a (Chl-a) concentrations (dominated by smaller cells) varied little through
time. Contrary to our expectations, even though $NCP_{O2/Ar}$ and $NCP_{POC}$ rates agreed well, we also
observed significant discrepancies between $GPP_{O2/Ar}$ and $GPP_{POC}$, and different community
respiration rates ($CR_{O2/Ar}$ and $CR_{POC}$) during drifter period 2. The contrasting properties between
the two drifter deployments enable us to examine the coupling of $O_2$ and POC dynamics under
different ecological states, with implications for the use of $\Delta O_2/Ar$ and POC measurements as
proxies for GPP and NCP.

## 4.1 Decoupling of $O_2$ and POC dynamics in the mixed layer


**4.1.1. Drifter 1.** In the absence of significant POC sinking and net loss to the dissolved organic carbon (DOC) pool, POC-based productivity rates should approximate $O_2/Ar$-based rates (Claustre et al., 2008; White et al., 2017). However, at drifter station 1, both $GPP_{O2/Ar}$ and $NCP_{O2/Ar}$ greatly exceeded $GPP_{POC}$ and $NCP_{POC}$, respectively (Figs. 4, 5a; Table 1). Over the three successive 24-hour periods of drifter deployment 1, the absolute difference between GPP measures increased from 41 mmol C $m_{-2}$ $d_{-1}$ to 260 mmol C $m_{-2}$ $d_{-1}$, while the absolute difference between NCP estimates increased from 42 mmol C $m_{-2}$ $d_{-1}$ to 193 mmol C $m_{-2}$ $d_{-1}$. The NCP differences exceeded the propagated uncertainties in NCP during second and third days of the deployment. The transition to negative $NCP_{POC}$ values over the course of the drifter 1 deployment primarily reflected diminishing daytime rates of POC accumulation (dPOC/dt term in Eq. 8) (Fig. S5).

This apparent discrepancy between $NCP_{O2/Ar}$ and $NCP_{POC}$ is consistent regardless of the approach used to calculate NCP rates (Sect. 2.6.3, Table 2). However, comparisons of $\Delta O_2/Ar$-derived NCP relative to POC-derived NCP in 3-hour increments (Eq. 11) can reveal discrepancies on shorter time scales than daily-integrated values (Fig. 5c). Taken together, these GPP and NCP comparisons suggest that additional POC losses decoupled $O_2$ from POC dynamics during drifter period 1. While mixed layer $\Delta O_2/Ar$ was primarily impacted by the accumulation of $O_2$ from gross primary production (GPP) and $O_2$ loss from community respiration, diurnal variability in [POC] was likely affected by several additional loss factors, including particle export, photooxidation, grazing, and DOC production.

During a diatom bloom, enhanced aggregation of large silica-rich particles and zooplankton fecal pellet production can stimulate export of POC and diatom cells out of the mixed layer, progressively decreasing $NCP_{POC}$ relative to $NCP_{O2/Ar}$. A number of previous studies have reported enhanced particle fluxes associated with diatom blooms in various oceanic regions (Buesseler, 1998; Guidi et al., 2009; Brzezinski et al., 2015; Stukel et al., 2017) . The global compilation of Henson et al. (2012) reported maximum export fluxes of ~83 mmol C $m_{-2}$ $d_{-1}$ from Southern Ocean measurements, while Alkire et al. (2012) derived maximum export fluxes of 96 mmol C $m_{-2}$ $d_{-1}$ during termination of the North Atlantic spring bloom. Stukel et al.

(2017) applied the steady-state $^{234}$Th-$^{238}$U approach to quantify export fluxes of ~36 mmol C m$^{-2}$
d$^{-1}$ in the nearshore region of the Southern California Current system. The higher value estimates
are in the range of the discrepancy we observed between NCP$_{POC}$ and NCP$_{O2/Ar}$, suggesting that
POC export fluxes could potentially account for a significant fraction of the inferred POC loss at
drifter site 1.

Another likely POC loss is DOC production through cellular exudation, viral lysis and/or

grazing (Briggs et al., 2018; Claustre et al., 2008; Dall'Olmo et al., 2011; Lochte et al., 1993).
Loss of POC to the DOC pool would lower NCP$_{POC}$ without affecting NCP$_{O2/Ar}$ values if the
DOC produced is not respired in the mixed layer. While we did not conduct direct measurements
of DOC concentrations during the cruise, previous work in a variety of ocean environments has
shown that DOC production can account for 3-37% of NCP in the Ross Sea, up to 10-40% in the
equatorial Pacific Ocean, and up to 66% in the Sargasso Sea during the seasonal phytoplankton
bloom (Hansell and Carlson, 1998). More recently, Alkire et al. (2012) estimated that 22-40% of
NCP was released into the DOC pool during the North Atlantic bloom, and Bif and Hansell
(2019) estimated springtime ΔDOC/NCP ratios of 0.05 – 0.54 and summertime ratios of 0 – 0.28
along the Line P transect (130 – 152 °W) in the eastern Subarctic Pacific. In the results of Bif
and Hansell (2019), the most comparable Line P measurement to drifter station 1 (in terms of
location and [Chl-a]) exhibited a ΔDOC/NCP ratio of 0.19 in the summer and 0.34 in the spring,
implying that up to 34% of NCP was partitioned into the DOC pool. Assuming a lower bound of
~20% of NCP released as DOC yields a daily-integrated DOC flux of 21 to 33 mmol C m$^{-2}$ d$^{-1}$.
The remaining discrepancy between ΔO$_2$/Ar and POC-based NCP estimates (14 to 159 mmol C
m$^{-2}$ d$^{-1}$; average, 103 mmol C m$^{-2}$ d$^{-1}$) is potentially attributable to particle export. Taking an
upper bound of 40% of NCP as DOC production, which is closer to the easternmost station
sampled in Bif and Hansell (2019), yields a daily-integrated DOC flux of 56 to 67 mmol C m$^{-2}$ d$^{-}$
$^{1}$ (Fig. 4) and a residual export flux of -14 to 126 mmol C m$^{-2}$ d$^{-1}$ (average, 76 mmol C m$^{-2}$ d$^{-1}$)
(Table 1). This range of results demonstrate that DOC production cannot likely account for the
full discrepancy between ΔO$_2$/Ar and POC-based NCP estimates at drifter site 1, suggesting that
export fluxes are likely a significant mechanism for mixed layer POC loss.

A final consideration involves diurnal variation of zooplankton abundances and grazing

rates, which may introduce an additional POC loss process that contributes to decoupling
between mixed layer POC/C$_{ph}$ and dissolved ΔO$_2$ dynamics ( Dall'Olmo et al., 2011; Briggs et

al., 2018). During our expedition, we observed a strong signature of diel migrating zooplankton

based on increased night-time signal spikes in surface optical backscatter measurements (Burt

and Tortell, 2018). In addition to particle sinking and DOC excretion, these nighttime migrations

could enhance POC and $C_{ph}$ loss at night without depleting $\Delta O_2/Ar$, if POC uptake rates exceed

respiration rates. For example, (Wu et al., 2010) observed that mesozooplankton prefer to graze

diatom-dominated assemblages at night over day in the East China Sea. Assuming that biomass

accumulation rates from grazing surpasses grazer respiration rates (Dagg et al., 1982), these

diurnal variations would contribute to more POC loss than $O_2$ loss. In addition, once POC is

assimilated into the body of a grazer, it joins a larger particle size class that likely exceeds the

size-dependent detection limits of the beam attenuation coefficient (Stramski and Kiefer, 1991;

Marra, 2002; Claustre et al., 2008;), decreasing the $c_p$ signal used to derive POC.

**4.1.2 Sub-daily variations in community respiration.** Although $GPP_{O2/Ar}$ generally

exceeded $GPP_{POC}$ during drifter period 1, differences in $CR_{O2/Ar}$ and $CR_{POC}$ were smaller and not

statistically significant throughout the drifter period (Fig. 5b). During days two to three, the

$CR_{O2/Ar}$ values were larger than $CR_{POC}$ values. Thus, the discrepancies between $NCP_{O2/Ar}$ and

$NCP_{POC}$ (Sect. 4.1.1) may be attributed more to differences in gross accumulation of POC and $O_2$

(Claustre et al., 2008; White et al., 2017), rather than to differences in POC and $O_2$ losses. This

suggests that POC loss rates varied on sub-daily time scales through drifter deployment 1, and

were generally higher in the daytime than at night. This is supported by a weak correlation

between $\Delta O_2/Ar$-based and POC-based 3-hour NCP ($p<0.05$, $r_2= 0.39$) in Fig. 5c, which suggests

that the magnitude of decoupling between $\Delta O_2/Ar$ and POC dynamics varies throughout the day.

Indeed, the increasing discrepancy between $NCP_{O2/Ar}$ and $NCP_{POC}$ between days 1 and 3 of the

drifter period suggests increasing POC loss rates over this time frame.

In the dynamic, high productivity upwelling environment of drifter site 1, a number of

day/night variations in grazing rates, export fluxes, particle sinking velocities, DOC production

rates and mixed layer properties could lead to greater differences between $GPP_{O2/Ar}$ and $GPP_{POC}$

than between $CR_{O2/Ar}$ and $CR_{POC}$ (Waite and Nodder, 2001; Gernez et al., 2011; White et al.,

2017; Briggs et al., 2018). Lower nighttime grazing relative to daytime rates would diminish

$CR_{POC}$ relative to $CR_{O2/Ar}$ (White et al., 2017; Briggs et al., 2018). However, grazing is typically

more pronounced at night due to upward vertical migration of zooplankton (e.g., Burt and

Tortell, 2018). Mixed layer depth changes, particularly shoaling in the day and deepening at
night, can enhance POC export fluxes in the day and lower fluxes at night (Gardner et al., 1999;
Briggs et al., 2018). However, we did not observe any consistent diel patterns in mixed layer
depth at drifter station 1.

More likely, higher DOC production relative to DOC respiration in the day and vice-

versa at night would cause $GPP_{O2/Ar}$ to increase more than $GPP_{POC}$ in the daytime, while causing
$CR_{O2/Ar}$ to exceed $CR_{POC}$ rates in the nighttime. Such light-dependent increases in DOC
production, could result, for example, from the effects of photo-respiration and other
mechanisms of dissipating excess light energy. Indeed, we observed evidence of photo-oxidative
stress during the daytime from increased non-photochemical quenching activity in phytoplankton
assemblages (Schuback et al., 2019). Finally, growth in cell size during the day could lead to
higher daytime export rates, while also causing proportionally more particles to escape detection
by the ac-s sensor in the daytime relative to at night, when increased cell division and depression
of particle sizes could increase the number of particles detected by the ac-s sensor (DuRand and
Olson 1996; Oubelkheir and Sciandra, 2008; Khierrediene and Antoine, 2014). We did not find
consistent changes in cell size from the particulate backscatter time series or *in situ*
measurements of size-fractionated [Chl-a] between day and night. Nonetheless, without direct
measurements of particle size, export fluxes, or DOC production, we cannot rule out the
influence of diurnal variations in such rates on our productivity estimates.

Taken together, the potential variations in POC loss rates on sub-daily time scales suggest

that comparing only GPP or only CR estimates based on nighttime linear regressions of $\Delta O_2/Ar$
and POC against time (Eqs. 1, 8) could yield erroneous estimates of POC loss. By comparison,
differences in daily $NCP_{O2/Ar}$ and $NCP_{POC}$ provide a more robust indicator of cumulative POC
loss, as illustrated by the relatively consistent discrepancies between all calculated $NCP_{O2/Ar}$ and
$NCP_{POC}$ values in Table 2.

**4.1.3 Drifter 2.** Drifter site 2 exhibited comparable absolute discrepancies between

$GPP_{O2/Ar}$ and $GPP_{POC}$ and greater discrepancies between $CR_{O2/Ar}$ and $CR_{POC}$ relative to drifter
site 1 (Fig. 5a-b; Table 1). A scatterplot of $\Delta O_2/Ar$-derived NCP and net [POC] change in 3-hour
increments (Eq. 11) shows that the magnitude of $\Delta O_2/Ar$-derived changes consistently exceeds
the magnitude of POC-derived changes throughout the drifter period, no matter the time of day
(Fig. 5d). The strong, positive relationship between these two 3-hour measures ($p<0.05$, $r^2=0.64$),
compared to the weaker correlation at drifter site 1 (Fig. 5c), suggests that despite large
differences in the magnitude of $\Delta O_2/Ar$-derived and POC-derived GPP and CR rates, POC-based
changes are a good relative indicator of $O_2$-derived productivity rates.

Prior studies have observed that the amplitude of diurnal variability in $\Delta O_2/Ar$ exceeds

the amplitude of diurnal variability in $c_P$-based [POC]. For example, in their North Atlantic
bloom survey, Briggs et al. (2018) observed higher amplitude variations in $O_2$ relative to $c_P$-
derived [POC], leading to higher absolute $O_2$-derived respiration and gross oxygen production
(GOP) rates compared to $c_P$-derived rates throughout stages of the bloom. A photosynthetic
quotient (PQ) of 1.45 (from Laws 1991) was not sufficient to reconcile GOP with $GPP_{cp}$. In the
Southern Ocean, Hamme et al. (2012) also observed high ratios of underway $\Delta O_2/Ar$-derived
gross oxygen production to gross carbon production (i.e., GPP) based on photosynthesis-
irradiance incubations, surpassing the expected range for the photosynthetic quotient. At a
relatively low productivity site with low phytoplankton biomass (Table 1; Fig. 2), heterotrophic
bacteria can comprise a substantial fraction of total living biomass in the mixed layer, and
variations in their total biomass can impact $c_P$ measurements (Oubelkheir and Sciandra, 2008;
Barnes and Antoine, 2014). If detected by the ac-s sensor, bacteria could potentially account for
some of the discrepancy between diel POC and $O_2$-derived variability at drifter site 2. Assuming
that DOC exudation from phytoplankton cells is positively related to growth in heterotrophic
biomass, either from direct DOC consumption, or indirectly through external drivers such as
irradiance levels (Kuipers et al., 2000; Fuhrman et al., 1985; Church et al., 2004; Oubelkheir and
Sciandra, 2008), $c_P$ decreases from phytoplankton exudation would counter $c_P$ increases from
heterotrophic growth. At night, this would decrease $CR_{POC}$ rates derived from $c_P$-based [POC],
relative to $O_2$-derived CR rates. Indeed, the positive $CR_{O_2/Ar}$ - $CR_{POC}$ discrepancy contributed to
58-82% of the differences between $\Delta O_2/Ar$ and POC-derived GPP rates at drifter station 2. The
remaining difference may be attributed to POC losses to the DOC pool or other sinks (discussed
below) in the daytime.

Because daytime increases in both $\Delta O_2/Ar$ and [POC] are balanced by respective

nighttime decreases, absolute differences in $NCP_{O_2/Ar}$ and $NCP_{POC}$ were smaller than at drifter
site 1. While, this discrepancy was negligible over the first 24-hour period, it increased to 32
mmol C m$_{-2}$d$_{-1}$ over the 24-hour period (Table 1; Fig. 4), exceeding the uncertainty of both NCP
calculations. Overall, the closer absolute agreement across NCP estimates is consistent with the
view of drifter site 2 as a more oligotrophic ecosystem, where primary production and
heterotrophic consumption are more tightly coupled (Claustre et al., 2008; White et al., 2017).
The smaller absolute differences between $NCP_{O2/Ar}$ and $NCP_{POC}$ suggest a lower, but non-
negligible, potential for POC sinking, grazing and net DOC production over consumption to
decouple POC, $C_{ph}$ and $\Delta O_2/Ar$ dynamics at drifter site 2. Although we lack direct DOC
measurements, lower 440 nm absorption values in the filtration blanks (Sect. 2.2) at drifter site 2
compared to drifter site 1 suggest lower colored dissolved organic matter (CDOM)
concentrations (Organelli et al., 2014; Peacock et al., 2014). This observation is consistent with
several previous observations of lower net DOC production in lower productivity and/or
oligotrophic waters (Bif et al., 2018; Hansell and Carlson, 1998). A recent compilation of
summertime DOC production and NCP measurements along the Line P transect in the Northeast
Pacific Ocean, shows that DOC production comprises at most 28% of total NCP in offshore
waters (Bif and Hansell, 2019). Even DOC/NCP ratios as high as 28% at drifter site 2 would
result in low overall DOC accumulation, because NCP rates were relatively low.

Low particle sinking rates are another factor that can explain the smaller absolute

discrepancy between $NCP_{O2/Ar}$ and $NCP_{POC}$ at drifter site 2. Low particle export is generally
expected from phytoplankton assemblages dominated by small particle sizes <20µm, consistent
with the higher $b_{bp}$ slope values and Chl-a size fractionation measurements at drifter site 2 (Sect.
3.2; Fig. 2) (Fowler and Knauer, 1986; Guidi et al., 2008). Nonetheless, POC export does occur
under low productivity conditions, and even small export fluxes could account for the entire
discrepancy between measures of NCP at drifter site 2. For example, Durkin et al., (2015)
reported significant rates of particle sinking from the small-celled, oligotrophic communities that
dominate the BATS station. In addition, it is possible that grazing by zooplankton would also
enhance loss of these phytoplankton cells from the mixed layer (Guidi et al., 2009). As we
observed at drifter site 1, increased variability in the $b_{bp}$ signal suggest the presence of vertically
migrating zooplankton into the mixed layer during nighttime intervals of drifter period 2 (Burt
and Tortell, 2018). Assuming that a maximal fraction of 28% of $NCP_{O2/Ar}$ is DOC production at
drifter site 2 (Bif and Hansell, 2019), a residual POC export flux of 23 mmol C m$^{-2}$ d$^{-1}$ would be
necessary to balance $NCP_{O2/Ar}$ and $NCP_{POC}$ during day two of the drifter period (Table 1). This
value is reasonable considering previous estimates reported from a number of lower productivity
systems (Henson et al., 2012; Charette et al., 1999).

**4.2 Other factors driving variability in NCP$_{POC}$**

In interpreting our results, it is critical to consider a number of potential caveats,

including methodological uncertainties and other POC sinks that could contribute to the
variability in derived NCP estimates, POC export and DOC excretion rates. One important
variable in all of our comparisons of productivity rates derived from biological oxygen saturation
and POC is the $O_2$-to-POC conversion factor, represented by the photosynthetic quotient (PQ)
value selected for each drifter site. Neglecting to take different respiratory quotients (RQs) into
account in this $O_2$-to-POC conversion (e.g., Ferrón et al., 2015) may contribute to uncertainty in
calculated GPP, CR and NCP rates, leading to erroneous discrepancies among derived values.
But, given the relatively narrow range (~50%) of possible PQs and RQs applicable to our study
sites (Laws 1991), a different PQ or RQ cannot account for the total discrepancy observed
among $\Delta O_2/Ar$ and POC-derived GPP, CR and NCP rates.

In our analysis, we interpret variations in particulate backscatter ($b_{bp}$) and beam

attenuation ($c_p$) in terms of phytoplankton and total particulate organic carbon concentrations,
assuming a small influence of inorganic suspended minerals from the continental shelf,
Columbia River discharge or other sources. However, the Columbia River plume has been
observed to extend south along the coast as far as ~44.5° N in the summertime (Thomas and
Weatherbee, 2006), close to the location of drifter deployment 1. Moreover, the drifter was
deployed ~40 km from shore over the continental shelf, where bottom resuspension of particles
and their subsequent upwelling into the mixed layer is possible. Estimates of the bulk refractive
index of particles ($\eta_p$), can be used to estimate the influence of inorganic minerals in our optical
measurements. During drifter deployment 1, we observed median $\eta_p$ values at 470, 532 and 650
nm that were generally below 1.12 (Fig. S3e), whereas inorganic minerals in seawater, have a
bulk refractive index as high as 1.26 (Lide, 1997; Twardowski et al., 2001). In addition, mixing
with the fresh Columbia River plume would have significantly reduced salinity at drifter site 1 to
values below 30 g/kg (Hickey et al., 1998), well below the 32 g/kg we observed during this
drifter deployment (Sect. 3.1; Fig. S3c), which are consistent with salinities observed in the
offshore Northeast Pacific Ocean (Whitney and Freeland, 1999). While these relatively high
salinities support our assertion of a negligible influence of riverine particles on our
measurements, the observed $\eta_P$ values do not preclude the presence of mixing between POC and
a small fraction of shelf-derived inorganic particles at drifter site 1. By contrast, calculated $\eta_P$
values during deployment 2 were below 1.08, which is close to values expected for water-
containing predominantly non-diatom phytoplankton organic carbon.

Additional uncertainty in our analysis derives from the algorithms used to estimate POC

and phytoplankton carbon $C_{ph}$ from optical measurements (Sect. 2.2). Because of particle size
limitations in the optical measurements, variability in seawater optical properties may not fully
capture all significant components of the particulate pool, such as larger microplankton and
zooplankton. Indeed, larger zooplankton often appear as erratic signal spikes in backscatter data
(Burt and Tortell, 2018), which are typically filtered out during data processing. Moreover, the $c_P$
signal at 660 nm, used to derive [POC], responds most strongly to particles within the 0.5–20 µm
diameter range (Claustre et al., 2008; Marra, 2002; Stramski and Kiefer, 1991), which is smaller
than many large diatoms, fecal pellets and particle aggregates. This size bias would cause an
underestimate of larger particles, and therefore [POC], measured by beam attenuation, thereby
contributing to the apparent discrepancy between diel changes in [POC] and diel changes in
$\Delta O_2/Ar$ (Fig. 4). Despite these potential caveats, recent work ( Graff et al., 2016; Briggs et al.,
2018; Burt et al., 2018) has demonstrated that $c_P$ and $b_{bp}$-based derivations of [POC] and [$C_{ph}$]
can indeed be robust in high biomass ocean regions, where productivity and the proportion of
large-celled phytoplankton may be greater.

Changes in the $c_P$-to-[POC] relationship through time could also drive apparent

variability in our optical [POC] estimates during both drifter deployments. On a global scale, the
linear regression of [POC] against $c_P$ at 660 nm measured in samples from diverse marine
environments is defined over a range of POC concentrations from ~5 to ~175 µg/L (Graff et al.
2015). At drifter site 2, the POC concentrations fell within the range of this fit. The assumption
of a constant POC/$c_{p660}$ ratio close to the value suggested by Graff et al. (2015), is less likely to
impact the derivation of apparent POC standing stocks and associated NCP estimates. Based on
relatively small changes in $b_{bp}$ slope values (Figs. S3d, S5d) and phytoplankton community
composition, it is unlikely that changes in particle size and bulk refractive index would have
significantly shifted the relationship between POC and $c_{p660}$ during drifter deployment 2.
As concentrations of POC at drifter station 1 were 25% higher than the empirical limits
of the $c_p$-based algorithm in (Graff et al., 2015), a different POC/$c_p$ relationship (i.e., different
slope of the linear fit) could apply. In a limited comparison with discrete POC samples, we found
a POC–$c_p$ slope that was similar to that of Graff et al. (albeit with a different y intercept) (Fig.
S2). Nonetheless, we cannot rule out changes in the $c_{p660}$–[POC] relationship due to shifts in cell
size and, to a lesser extent, bulk refractive index resulting from diatom accumulation
(Kheireddine and Antoine, 2014; Stramski and Reynolds, 1993) (Fig. S3d–e). Indeed, Briggs et
al. (2018) observed that the ratio of [POC] to $c_p$ decreased by ~20% during the rise of the North
Atlantic bloom, while values increased by ~60% during the bloom decline. If we assume a 20%
decrease in POC/$c_{p660}$ values (from ~420 to ~340 mg m$_{-2}$) associated with diatom growth (Briggs
et al., 2018), our daily NCP$_{POC}$ estimates would be closer to 0, less positive during day 1 and less
negative during days 2–3. This, in turn, would increase the apparent decoupling between NCP$_{POC}$
and NCP$_{O2/Ar}$ on days one (~27%) and three (~1%), and bring the values slightly closer on day
two (~8%). Overall, the value of these potential changes is small relative to the differences we
observed between NCP$_{O2/Ar}$ and NCP$_{POC}$, and we thus conclude that variable POC/$c_{p660}$ ratios
cannot explain the observed decoupling between POC, $C_{ph}$ and dissolved $O_2$ dynamics at the
drifter 1 site.
Finally, error associated with the POC mixing correction could affect calculated NCP$_{POC}$
values (Eq. 8) and therefore the discrepancy between NCP$_{O2/Ar}$ and NCP$_{POC}$, and derived export
estimates. This vertical mixing correction for NCP$_{POC}$ is based on average parameters derived
from N$_2$O measurements for the whole drifter period (Sect. 2.5). This introduces some error in
day-to-day corrections to the NCP$_{POC}$ calculations. In addition, the gradient term dPOC/dz in Eq.
10 is based on the difference between average POC concentrations measured at two depths
during CTD deployments (5 m and one depth over 40-60 m). Because high-resolution
transmissivity profiles showed that particle concentrations reached a steady minimum between
30 m and 40 m in most CTD deployments, dz in Eq. 10 was taken as the difference between the
drifter 1 $z_{mld}$ and this daily average depth of minimum transmissivity, rather than the deeper POC
sampling depth (i.e., 40 – 60 m). Because variations in transmissivity do not necessarily equate
to variations in [POC], errors in dz would impact the vertical mixing correction and therefore
calculated NCP$_{POC}$ values. For example, if the [POC] minimum was actually deeper, this would
increase the value of dz and decrease dPOC/dz and the total mixing correction, yielding lower

933 NCP$_{POC}$ values and a higher discrepancy between NCP measures. In propagating the error for

934 NCP$_{POC}$, we have included the standard deviation of the minimum transmissivity depth across

935 daily CTD casts, which partially addresses this uncertainty in the dz term. Fortunately, the

936 NCP$_{POC}$ mixing corrections over drifter period 1 approximate the magnitude of the NCP$_{O2/Ar}$

937 mixing correction (Sect. 3.3, Table 1), increasing our confidence in the POC mixing correction

938 applied here.

939   Aside from uncertainties that directly impact estimates of NCP, there are a number of

940 other potential caveats in our analysis of phytoplankton carbon from b$_{bp}$ and particle size

941 distribution from b$_{bp}$ slope. Previous studies have reported that daily variations in b$_{bp}$ do not

942 always track daily variations in c$_p$, suggesting that b$_{bp}$ dynamics do not reflect phytoplankton

943 carbon dynamics on diel time scales (Kheireddine and Antoine, 2014; Briggs et al. 2018). We

944 observed a similar decoupling between b$_{bp}$ and c$_p$ in this study; for example, while c$_p$ values at

945 660 nm steadily declined in the last 24 hours of drifter period 1, b$_{bp}$ at 470 nm stayed relatively

946 constant. Nonetheless, [C$_{ph}$] estimates from b$_{bp}$ (Fig. 2) remain useful for comparisons between

947 drifter sites, and differences in apparent phytoplankton biomass concentration were consistent

948 with a number of the other biogeochemical differences measured between the two trophic

949 regimes. Similarly, the relationship between b$_{bp}$ slope and particle size distribution has been

950 challenged in previous literature (e.g., Zeng et al., 2018). While this limits our interpretation of

951 daily b$_{bp}$ slope dynamics, we did find independent evidence for larger particle sizes at drifter site

952 1 (as predicted by the b$_{bp}$ slope), from size fractionated Chl-a measurements and pigment

953 analysis showing a greater fraction of diatoms (Sect. 3.2).

955 **4.3 Reconciling NCP and NPP**

957   During both drifter surveys, we estimated daily-integrated net primary productivity

958 (NPP) values using carbon-based productivity model (CbPM) calculations and $_{14}$C bottle

959 incubations (Sect. 2.5). On several days, these two measures of NPP estimates were consistently

960 lower than NCP$_{O2/Ar}$ integrated over the same time scales and mixed layer depths (Table 1; Fig.

961 4). Similarly, Briggs et al. (2018) and Alkire et al. (2012) also reported NCP values that were

962 equal to or greater than NPP values obtained from different methodologies during their

963 Lagrangian study of the North Atlantic Bloom.

In theory, NCP cannot exceed NPP, as NCP includes additional respiration terms not
included in NPP, and must always be equal to or (more realistically) lower than NPP. Recent
work in the Northeast Pacific Ocean, has reported mean NCP/NPP ratios, based on $\Delta O_2/Ar$
measurements and CbPM calculations, in the 0.16 to 0.26 range for offshore and coastal waters,
respectively (Burt et al., 2018). These values, determined from continuous observations along a
moving ship-track are consistent with theoretical expectations. The observed high (>1) apparent
NCP/NPP values observed in our study and that of Briggs et al. (2018) and Alkire et al. (2012)
highlight a number of methodological limitations that could depress NPP estimates.
One possibility, which has been discussed at length by various authors (Gieskes et al.,
1979; Fogg and Calvario-Martinez, 1989; Marra, 2009), is that bottle containment effects limit
accurate estimates of $14C$ uptake. This likely caused underestimates of $14C$-NPP during both
drifter surveys, relative to CbPM-NPP and $NCP_{O2/Ar}$, which do not require discrete sample
incubations. In addition, during this last $14C$-uptake experiment of drifter survey 2, the incubator
warmed, which could have significantly impacted phytoplankton growth rates during the
incubation and result in depressed $14C$-NPP values, if thermal optima were exceeded.
A number of factors may also depress CbPM-based NPP estimates. While the model
applies a satellite-based relationship between $[Chl-a]/[C_{ph}]$ and daily mixed layer irradiance ($E_g$)
to calculate growth rate, these $E_g$ values may not fully parametrize phytoplankton physiology for
mixed assemblages in the ocean (Westberry et al., 2008). Indeed, phytoplankton
photophysiology varies with other environmental conditions and phytoplankton composition
(Cloern et al., 1995; Geider et al., 1998; MacIntyre et al., 2002; Westberry et al., 2008). In
addition, the CbPM does not allow calculated growth rates to exceed 2 d$_{-1}$, which may not apply
to all ocean environments (Graff et al., 2016). These uncertainties could potentially impact the
applicability of the CbPM parameters to the specific ocean conditions at drifter sites 1 and 2. In
addition, a vertical mixing correction for ac-s and backscatter-derived [Chl-a] and [$C_{ph}$],
respectively, not feasible in the present data set, may improve CbPM-based estimates of NPP.

**4.4 Comparison to other studies**

A number of previous studies have examined diurnal variation in upper ocean
phytoplankton and organic particle dynamics across a variety of productivity regimes, from
oligotrophic environments (Claustre et al., 1999, 2008; Wu et al., 2010; Gernez et al., 2011;
Kheireddine and Antoine, 2014; Thyssen et al., 2014; Nicholson et al., 2015; Ribalet et al., 2015;
White et al., 2017), to higher productivity waters and phytoplankton blooms (Brunet and Lizon,
2003; Wu et al., 2010; Alkire et al., 2012; Gernez et al., 2011; Dugenne et al., 2014; Kheireddine
and Antoine, 2014; Needham and Fuhrman, 2016; Briggs et al., 2018). In general, these studies
have shown that more productive environments exhibit higher amplitude diurnal variations in
beam attenuation, POC concentration, phytoplankton cell abundances, Chl-a, and metabolic
rates, as compared to oligotrophic regions. These prior results are consistent with the differences
we observed between the two distinct Northeast Pacific trophic environments represented by
drifter sites 1 and 2, respectively (Sect. 3.2; Figs. 2, S5).

To our knowledge, however, only two previous studies have directly compared diurnal

variations in $O_2$-based and $c_P$-based mixed layer productivity using Lagrangian drifters (Alkire et
al., 2012; Briggs et al., 2018). This previous work demonstrated that GPP and NCP dynamics
derived from dissolved $O_2$ measurements differed from net POC accumulation over the course of
the North Atlantic bloom, with the magnitude of this disparity varying as a function of bloom
stage. The authors found that highest rates of POC export and DOC production, corresponding to
the greatest $O_2$-POC discrepancy, occurred during the main period of the bloom development,
prior to its termination. The results of our study off the Oregon coast extend these previous
observations from the North Atlantic bloom into two new surface ocean regimes: a high
productivity Pacific upwelling zone, and a lower productivity offshore region. The upwelling
environment was characterized by rapid diatom accumulation, yielding significant differences
between $NCP_{O2/Ar}$ and $NCP_{POC}$, and $GPP_{O2/Ar}$ and $GPP_{POC}$. We also observed significant
differences between $\Delta O_2$/Ar-based and POC-based GPP and CR rates at the lower productivity
drifter 2 site, even though daily-integrated measures of NCP and net carbon accumulation agreed
more closely.

While most previous work across oligotrophic environments has highlighted the

agreement between GPP derived from daily variability in beam attenuation and dissolved $O_2$
(e.g., Claustre et al., 2008; White et al., 2017), our results illustrate two different examples where
$\Delta O_2$/Ar-based and POC-based GPP rates do not agree. We have found that even lower
productivity environments like drifter site 2 can display a quantifiable discrepancy between
productivity measures. At this site, even though POC-derived GPP and CR consistently
underestimated $\Delta O_2$/Ar-derived rates, net changes in [POC] were a sufficient relative indicator of
variations in $\Delta O_2$/Ar-based productivity, as has been observed in previous work (Briggs et al.,
2018). As a result, NCP measures agreed well, supporting the continued use of diurnal
measurements of beam attenuation to estimate NCP$_{POC}$ in low productivity regimes, where POC
and $O_2$ dynamics are closely coupled. In higher productivity regions like at drifter site 1 or the
area of the North Atlantic Bloom (Alkire et al., 2012; Briggs et al., 2018), measurements of both
POC and $O_2$ are likely required to constrain organic carbon mass balance, where POC and $O_2$
dynamics can be significantly uncoupled on short time scales. Measurements that simultaneously
estimate surface water $O_2$ accumulation, net DOC production and vertical transport of deep water
to the mixed layer at high temporal resolution offer the opportunity to evaluate the fate of NCP.
These quantities are especially important in the California coastal upwelling regime and other
similar ecosystems, with high NCP and significant potential for carbon transfer to higher trophic
levels.

**5 Conclusions**

In the current study, biological oxygen saturation ($\Delta O_2$/Ar) and optically-derived
particulate organic carbon (POC) were measured continuously and simultaneously during two
Lagrangian drifter deployments. This dual measurement approach allowed us to examine the
(de)coupling between carbon and dissolved oxygen in surface waters, and facilitated direct
comparison of $O_2$/Ar and POC-derived measures of gross primary productivity (GPP),
community respiration (CR), and net community production (NCP), from a mesotrophic
upwelling-influenced system to a more oligotrophic system further offshore. As hypothesized,
the results show that $O_2$ and POC-based measures of GPP and NCP diverge in mid-to-high
productivity phytoplankton communities, where daily fluctuations in $\Delta O_2$/Ar are decoupled from
POC cycling. Interestingly, oxygen-based GPP and CR exceeded POC-based GPP and CR rates
at the lower productivity site too, though we found that net changes in POC scaled with changing
productivity based on $\Delta O_2$/Ar. Thus, NCP estimates at drifter site 2 showed better agreement
because $O_2$ and POC cycles appeared to be more tightly coupled.
These findings are generally consistent with current understanding of productivity
dynamics and mixed layer POC cycling in these two coastal Pacific environments, and
complement only one prior comparison of daily GPP and NCP estimates from simultaneous,
autonomous measurements of $c_P$ and $O_2$ in the North Atlantic mixed layer (Alkire et al., 2012;
Briggs et al., 2018). Importantly, however, our results differ from earlier studies by providing
two examples of significant disagreement between $GPP_{O2/Ar}$ and $GPP_{POC}$, and $CR_{O2/Ar}$ and $CR_{POC}$
rates. We have further shown that for upwelling regions like drifter site 1, it is important to
account for vertical mixing of sub-surface waters into the mixed layer, and its effect on not only
$NCP_{O2/Ar}$ calculations (Izett et al., 2018), but also on $NCP_{POC}$ estimates through dilution of the
surface POC signature. Thus, our study illustrates an application of the vertical mixing
coefficient, $k_{mix}$, derived from [N$_2$O] profiles, to more accurately estimate net changes in POC
and nutrient concentration in such environments.

Moving forward, the disparity between POC and $O_2$-based NCP estimates offers an

opportunity to continuously track cumulative POC losses in the mixed layer using autonomous
ship-board or in situ sensors. The results show that this approach performs well in distinguishing
regions of high particle export, notwithstanding some major methodological limitations (Sect.
4.2) and poorly constrained DOC production rates (Sect. 4.1.1), which increase the uncertainty of
our export estimates at drifter site 1. As it is difficult and labor intensive to measure POC export
on short time scales with sediment traps and the $_{234}$Th-$_{238}$U disequilibrium method (Buesseler et
al., 2006; Savoye et al., 2006), simultaneous underway measurements of dissolved $O_2$,
particulate beam attenuation and CDOM absorption and spectral slope over a range of
wavelengths <400 nm (Del Vecchio and Blough, 2004; Grunert et al., 2018) may provide a
valuable, first-order approximation of POC partitioning among living phytoplankton biomass,
particle export and dissolved organic carbon (DOC) in the surface ocean on short time scales.

For future work, we recommend a number of approaches to increase our confidence in

derived POC export from coupled $O_2$, POC, and DOC dynamics. First, it will be valuable to
constrain particle size, and partitioning of POC into detrital and living (phytoplankton and
heterotrophic bacteria) components to properly assess the size range captured by optics-based
POC and $C_{ph}$ measurements. Second, independent estimates of POC export and DOC
concentrations during each drifter deployment could validate estimates of POC export fluxes
derived from coupled $O_2$ and POC measurements. Relatedly, depth-resolved backscatter profiles
(Briggs et al., 2013, 2018) could be used as another autonomous approach to calculating export
fluxes, as an independent check on surface-based estimates. Going forward, there is significant
future potential to exploit coupled $O_2$ and $c_p$ measurements on autonomous platforms, including
various ocean moorings (e.g., the Optical Dynamics Experiment, the Biowatt II program, and the
Bermuda Testbed Mooring program), and biogeochemical floats and gliders to resolve
opportunistic, high-resolution POC export time series (Stramska and Dickey, 1992; Kinkade et
al., 1999; Dickey and Chang, 2002). Deployment of such autonomous measurement systems
across a range of oceanic regions will help to constrain POC and productivity dynamics on
global scales.

**Data availability**

Discrete and underway optical measurements may be accessed at
https://github.com/srosengard/rosengard-tortell-oc2017.git

**Author contributions**

Sarah Rosengard, Philippe Tortell, and Nina Schuback collected the data in the field. Robert Izett
processed the CTD cast data and nitrous oxide measurements. Sarah Rosengard wrote the
manuscript with significant input from the co-authors.

**Competing interests**

The authors declare that they have no conflict of interest.

**Acknowledgements**

Special thanks to Jessie Gwinn, Jay Pinckney, Ross McCulloch, Chen Zeng, Melissa Beaulac,
Chris Payne and Maureen Soon for assistance in field collection and analysis of samples, and to
two anonymous reviewers for greatly strengthening the interpretations in this manuscript. This
project was funded by the Natural Sciences and Engineering Research Council of Canada
(NSERC), and by the US National Science Foundation (NSF project number 1436344).

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

**Table 1**: Daily-integrated mixed layer net primary production (NPP) and net community production (NCP), including all components used to calculate NCP using $\Delta O_2/Ar$ or POC time series, as indicated: gross primary productivity (GPP), respiration (CR), vertical mixing (Mix), and gas exchange ($J_{ex}$). Derived POC export estimates assuming a maximum of 40% and 28% DOC/NCP$_{O2/Ar}$ during drifter periods 1 and 2, respectively, are provided, as well (Sects. 4.1.1, 4.1.3). All units here are in mmol C m$^{-2}$ d$^{-1}$. Note that CbPM is the carbon-based production model (Sect. 2.5).

| | Drifter 1: | | | Drifter 2: | |
|---|---|---|---|---|---|
| | Day 1 | Day 2 | Day 3 | Day 1 | Day 2 |
| **NPP (CbPM)** | $147 \pm 61$ | $137 \pm 51$ | $112 \pm 40$ | $22 \pm 9$ | $18 \pm 7$ |
| **NPP ($_{14}$C)** | $150 \pm 18$ | - | $49 \pm 8$ | $12 \pm 4$ | - |
| **GPP ($\Delta O_2/Ar$)** | $284 \pm 75$ | $270 \pm 178$ | $358 \pm 198$ | $108 \pm 101$ | $219 \pm 211$ |
| **GPP (POC)** | $242 \pm 51$ | $106 \pm 26$ | $98 \pm 35$ | $41 \pm 8$ | $38 \pm 7$ |
| **R ($\Delta O_2/Ar$)** | $-73 \pm 65$ | $-150 \pm 88$ | $-172 \pm 56$ | $-83 \pm 35$ | $-186 \pm 64$ |
| **R (POC)** | $-77 \pm 55$ | $-147 \pm 28$ | $-104 \pm 40$ | $-44 \pm 12$ | $-36 \pm 9$ |
| **Mix ($N_2O$)** | $70 \pm 29$ | $16 \pm 81$ | $19 \pm 42$ | $0$ | $0$ |
| **Mix (POC)** | $67 \pm 47$ | $12 \pm 16$ | $20 \pm 16$ | $0$ | $0$ |
| **$J_{ex}$ (daily)** | $-62 \pm 11$ | $-7 \pm 4$ | $-6 \pm 3$ | $12 \pm 5$ | $17 \pm 7$ |
| **NCP$_{O2/Ar}$** | $140 \pm 45$ | $104 \pm 84$ | $167 \pm 52$ | $-12 \pm 44$* | $33 \pm 20$ |
| **NCP$_{POC}$** | $97 \pm 49$ | $-53 \pm 24$ | $-25 \pm 31$ | $-2 \pm 3$ | $1 \pm 2$ |
| **POC export** | $0$ | $115$ | $126$ | $0$ | $23$ |

*From three-hour increments of NCP$_{O2/Ar}$ (refer to Table 2). All other NCP values computed using day/night linear regressions of [POC] and [$O_2$]$_{bio}$ against time (Sects. 2.6.1, 2.6.2).

Table 2: Comparisons of NCP calculated using different time scales of integration (Sect. 2.6.3).
For every calculation approach, "Export + DOC" is the average difference between $NCP_{O2/Ar}$ and
$NCP_{POC}$ values ± 1 S.D. or ± the propagated error. All units here are in mmol C $m^{-2}$ $d^{-1}$.

| | Drifter 1: | | | | Export + DOC | Drifter 2: | | |
|---|---|---|---|---|---|---|---|---|
| | Day 1 | Day 2 | Day 3 | Mean ± S.D. | | Day 1 | Day 2 | Mean ± S.D. |
| $NCP_{O2/Ar}$ | 140 ± 45 | 104 ± 84 | 167 ± 52 | 137 ± 32 | | 26 ± 18 | 33 ± 20 | 29 ± 5 |
| $NCP_{POC}$ | 97 ± 49 | -53 ± 24 | -25 ± 31 | 7 ± 80 | 131 ± 79 | -2 ± 3 | 1 ± 2 | -0.8 ± 3 |
| $NCP_{O2/Ar}$ (3 hr) | 177 ± 121 | 129 ± 102 | 122 ± 157 | 143 ± 30 | | -12 ± 44 | 25 ± 75 | 6 ± 26 |
| $NCP_{POC}$ (3 hr) | 119 ± 66 | -86 ± 64 | 53 ± 140 | 28 ± 105 | 115 ± 88 | -8 ± 10 | -6 ± 5 | -7 ± 1 |
| $NCP_{O2/Ar}$ (time points) | 180 ± 54 | 128 ± 84 | 78 ± 43 | 129 ± 51 | | -4 ± 13 | 26 ± 11 | 11 ± 21 |
| $NCP_{POC}$ (time points) | 99 ± 48 | -73 ± 21 | -14 ± 19 | 4 ± 87 | 124 ± 66 | -6 ± 17 | -2 ± 11 | -4 ± 3 |
| $NCP_{O2/Ar}$ (whole drifter trend) | | | | 103 ± 56 | | | | 13 ± 9 |
| $NCP_{POC}$ (drifter trend) | | | | -21 ± 28 | 121 ± 76 | | | -4 ± 2 |


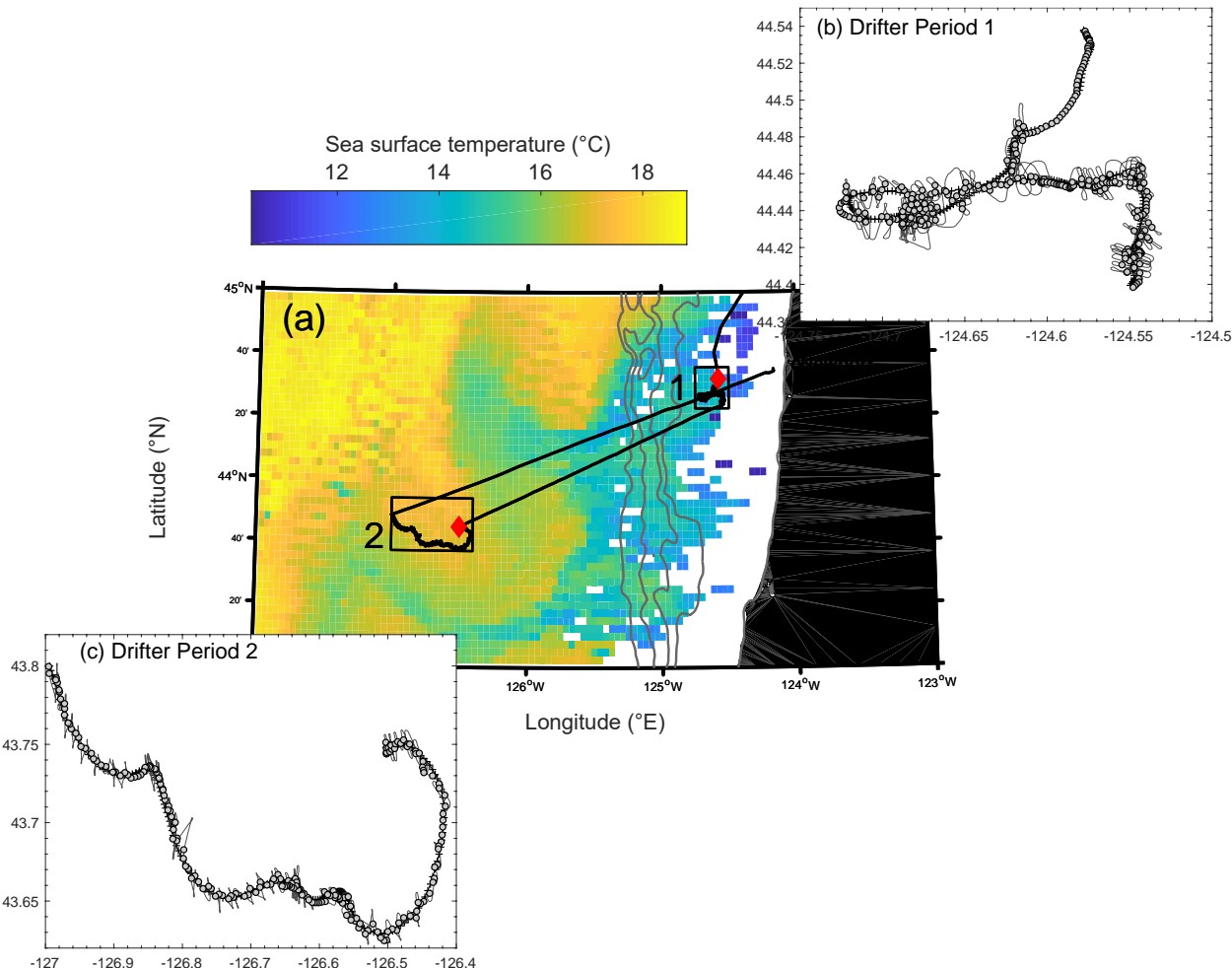


**Figure 1**: (a) Map of AQUA MODIS-derived 8-day composite sea surface temperature (11µm,
nighttime) from 21-28 August 2017, overlapping with the duration of both drifter deployments.
The two hollow boxes on the map denote location of drifter tracks, with the red diamonds
indicating the location of the initial release. Gray bathymetry contours extend from 0-2000 m,
with deepest contours representing the extent of the continental shelf. Panels (b and c) show a
detailed view of the two drifter tracks, with the ship's track shown in a light grey line and circles
denoting times when the ship was <1.5 km away from the drifter position. Only measurements
taken at these cross-over locations were used for analysis.

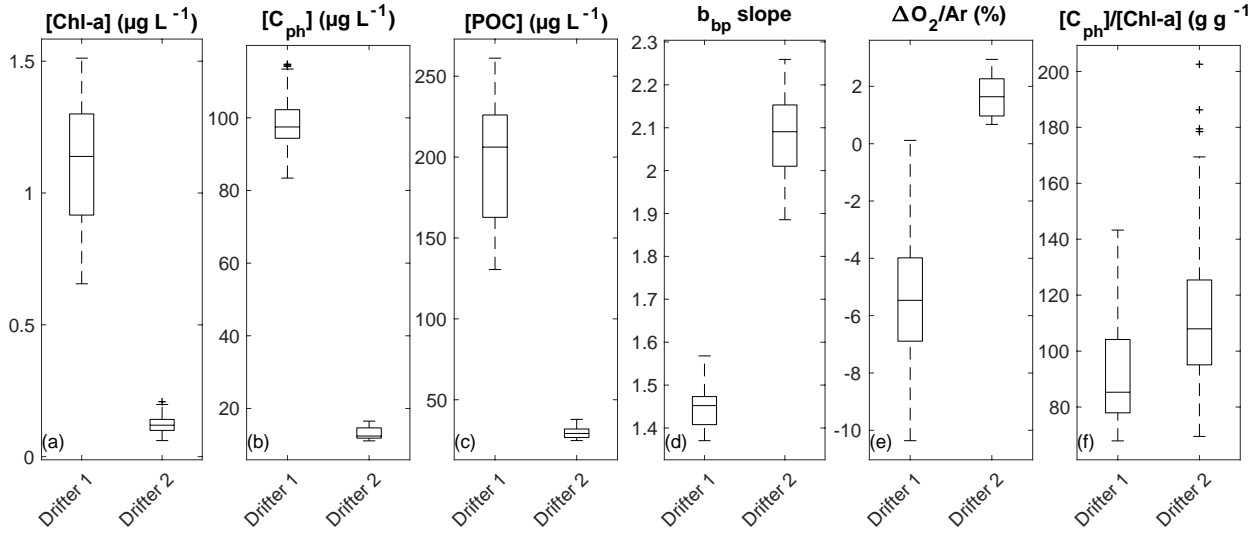


**Figure 2:** Comparison of average surface water properties between the two drifter deployments:
(a) chlorophyll-a concentration (Chl-a), (b) phytoplankton carbon concentration ($C_{ph}$), (c)
particulate organic carbon (POC) concentration, (d) the wavelength-dependent slope of
particulate backscatter ($b_{bp}$), (e) biological oxygen saturation anomaly ($\Delta O_2/Ar$), and (f) the
[$C_{ph}$]/[Chl-a] ratio. Boxes represent the median (center line) and 25 and 75 percentiles (box
edges). Outliers are indicated as black "+" marks.


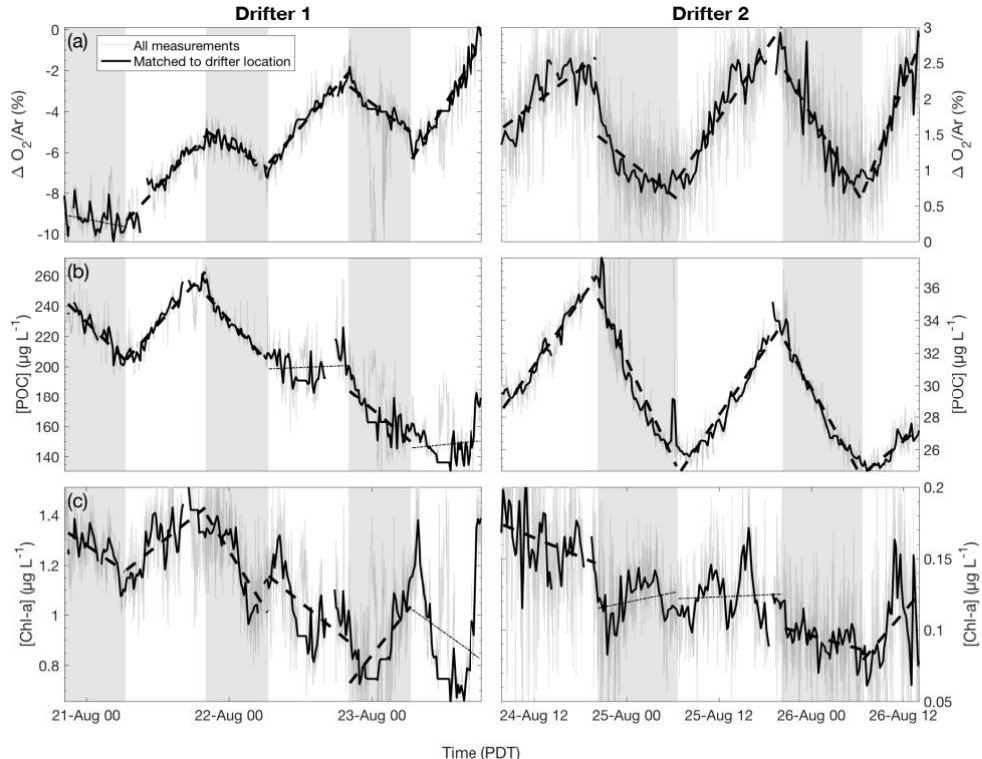


**Figure 3**: Time-series of (a) biological oxygen saturation ($\Delta O_2/Ar$), (b) particulate organic
carbon (POC) concentration, and (c) chlorophyll-a (Chl-a) concentration during the two drifter
deployments (left and right panels, respectively). For each daytime (non-shaded) and nighttime
(shaded) interval, the best fit linear regression line is plotted. Significant regressions ($p < 0.05$) are
plotted as thick dashed lines, while non-significant regressions ($p \geq 0.05$) are plotted as thin dotted
lines. Grey lines show all measurements while thicker black line shows observations collected
when the ship was within 1.5 km of the drifter location.

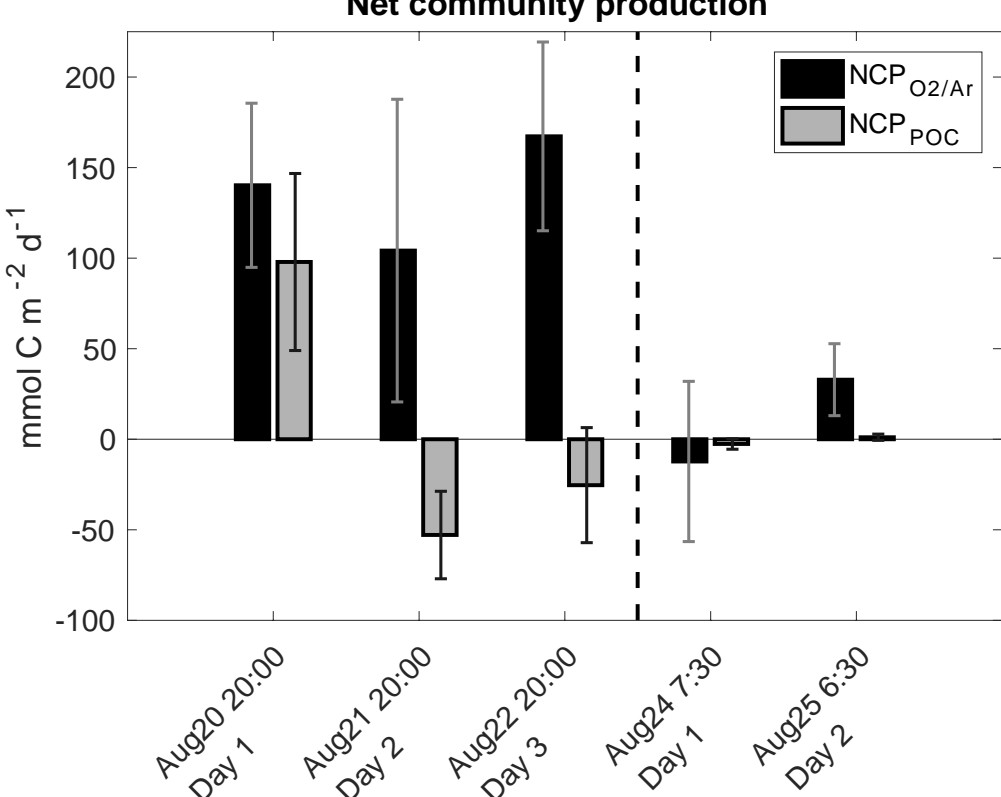

**Figure 4**: Daily net community production (NCP) during successive days of the two drifter
deployments derived from diurnal variations of biological oxygen saturation ($\Delta O_2/Ar$), and
particulate organic carbon (POC) concentration. Each set of bars is for one 24-hour period, with
approximate starting times on the x-axis. Note that the negative $NCP_{O2/Ar}$ value for the first day
of drifter period 2 was computed by integrating $NCP_{O2/Ar}$ values over eight consecutive three-
hour increments (refer to Table 2).

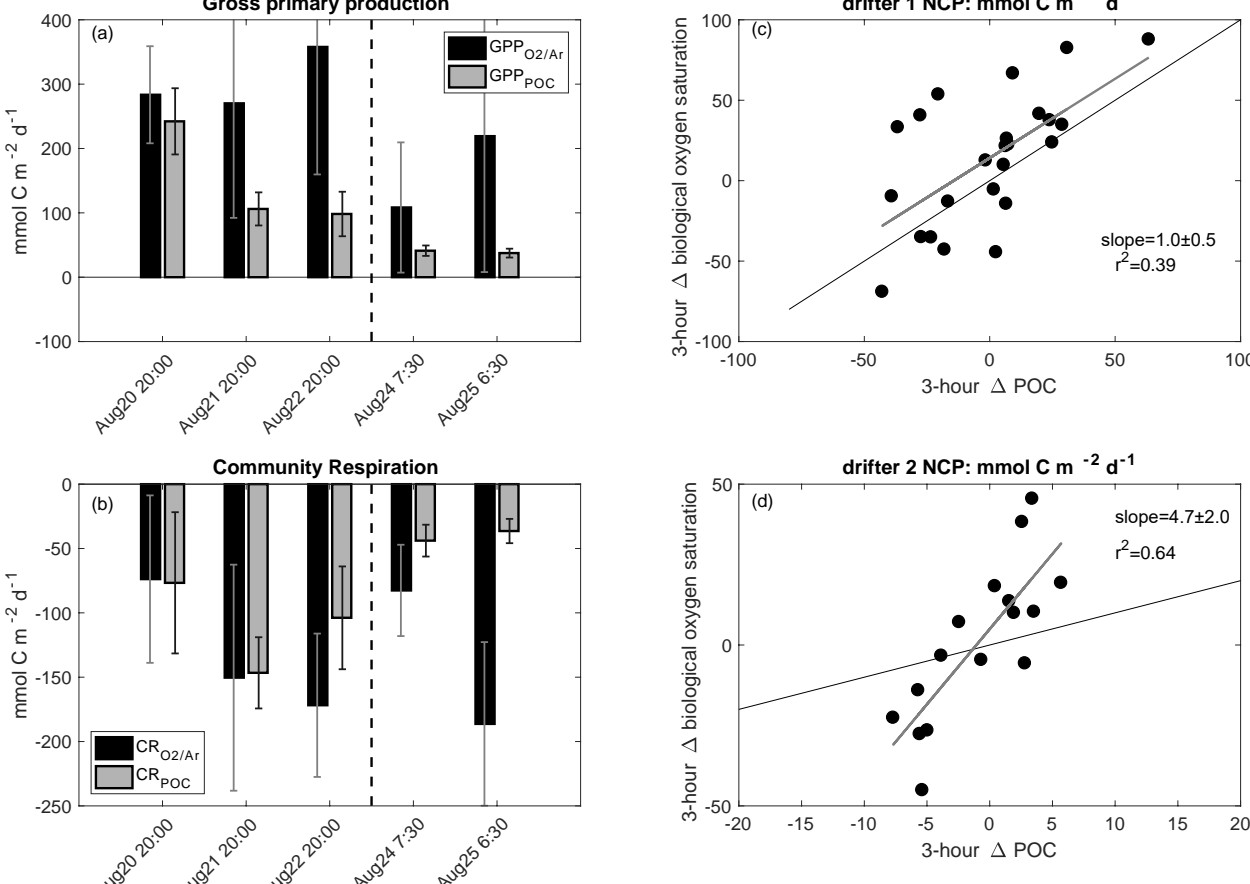

**Figure 5**: The left panels show comparisons between $\Delta O_2/Ar$-derived and POC-derived (a) GPP
and (b) CR over the five days of both drifter deployments. The right panels show $\Delta O_2/Ar$-derived
NCP (NCP$_{O2/Ar}$) as a function of POC-derived NCP (NCP$_{POC}$) over three-hour increments during
(c) drifter period 1 and (d) drifter period 2. Thin black lines in (c) and (d) represent the 1:1 line,
while thicker grey lines are the best-fit from linear regressions and correspond to the indicated
slope and $r_2$ values.