# Peer review of "Decoupling of $\Delta\text{O}_2/\text{Ar}$ and particulate organic carbon"

_Biogeosciences, 2019_

## Referee Comment (RC1) · Anonymous Referee #1 · 1 Aug 2019

**Summary**:

This manuscript analyzes two Lagrangian mixed-layer timeseries of particulate beam attenuation and biological oxygen saturation together with an array of complementary data in order to track the flow of organic carbon in two contrasting North Pacific environments. They find high net primary productivity (estimated via two methods) and net community productivity (estimated from biological oxygen saturation) in a coastal upwelling environment off the Oregon coast and much lower productivities further offshore. In both environments, NCP was similar to or higher than NPP estimates and substantially higher than net POC accumulation (calculated from changes in optical

beam attenuation). The difference between NCP and POC accumulation, combined with previous results on DOC release, implies that 13-45

**General Comments**:

For the most part, the text is clear, and the methodology appears thorough. I agree with the authors' general conclusion that, in principle, the combination of high-resolution POC (via cp) and O2/Ar timeseries is valuable for understanding carbon flows. However, I think that the manuscript needs to be better focused on clearer conclusions that arise directly from the results of this study. Put another way, after reading the manuscript, I am not sure exactly in what way the authors think that the specific results of this study have advanced scientific knowledge.

Overall, I recommend major revisions before publication. The interpretation of the various optical proxies should be tightened up, clarifying the extent of empirical support for each proxy, the use of the term "diel cycles" should be clarified, and the manuscript should be refocused around clear conclusions that stem directly from the results of this study. Also, please be specific about why the findings are important. For example, one of the conclusions seems to be that this method promises to expand the coverage of export estimates. How exactly (autonomous or ship-based), and what accuracy can we expect? This manuscript estimates export at 13-45

I see two main possibilities for the main conclusion of the manuscript (although I am open to others if they are clearly articulated and supported by the data):

1. Is the main advance a "proof of concept" of the approach (simultaneously tracking net O2 production and net POC accumulation to constrain export) in two additional environments? If so, what are the criteria for success of this "proof of concept"? What if all O2/Ar-NCP values had been 50

2. Is the main advance some new knowledge about the functioning of the specific two ecosystems studied (i.e. Upwelling and offshore N. Pacific)? If so, what exactly have

we learned and how does it differ from (or strengthen) previous understanding? To me, the only conclusion in the conclusions section that clearly comes from the data presented in this manuscript is that O2 and POC cycling is more "coupled" offshore than in the upwelling region. But to me this statement is vague; I don't understand exactly what it means or why it's important. It would be much clearer to say something like "we find that a higher fraction of production is exported in region X than region Y" or "O2/Ar-based NCP can be used as a proxy for carbon export in region X but not region Y". To me, it actually looks like O2-based NCP substantially exceeds POC accumulation in both environments. While the absolute difference is smaller in the low biomass region, it is not clear that the fraction of NCP that ends up as export and/or DOC is lower in the low biomass site.

Of course, the conclusions can be a combination of methodological validation (1) and oceanographic findings (2), but in any case, the conclusions need to be articulated clearly and directly connected to the results. Put another way, much of the conclusions section seems like it could have been written without seeing the results of this study. This makes the value of the work less clear.

In addition to the main conclusion(s), I think that some secondary conclusions could be better highlighted. For example, the authors tentatively conclude that their high O2/Ar-NCP estimates, if accurate, imply that both the widely-used CbPM model and the even more widely-used C-14 incubation method might be substantially under-estimating NPP in this environment. Even if this conclusion is not certain, if the authors believe that this is the most likely interpretation of their results, this conclusion is worth highlighting, because the accuracy of these NPP methods are of broad importance to the field.

Finally, in conjunction with improved focus on conclusions, I think that two analyses should be strengthened and clarified (POC and O2 Diel cycles, and nutrient drawdown) and three other analyses should be de-emphasized due to large uncertainties (phytoplankton carbon diel cycles, the CDOM-DOC connection, and the bbp spectral

slope-particle size connection). See details below.

**Analysis of "diel cycles":**

The authors refer many times in the manuscript to their analysis of "diel cycles". To me, "diel cycle" refers to the change in the balance of production and respiration over the course of a single day (esp. the difference between night and day rates of change). Analysis of diel cycles allows estimate of gross primary production. The authors cite many papers that calculate gross production or a related quantity from both O2 and/or cp diel cycles, but for some reason, despite independently quantifying night-time and daytime rates of change, the authors do not gross production of oxygen or carbon from these diel cycles. Instead, they add night and day gain/losses together to calculate NCP. For NCP calculation, diel cycles (sub-daily data) are not really needed at all. All you need is the net change from the beginning (or end) of one day to the next. So the repeated claim that diel cycles are used to calculate NCP does not make much sense to me. Diel-cycle-based gross production estimates have their own uncertainties, but I think that they would be very valuable for the interpretation of the results of this study. The authors hypothesize that both NPP methods may be biased low. A finding that diel-cycles-based gross production of O2 and POC are substantially higher than the NPP estimates would increase support for this hypothesis. A finding that gross production agrees with NPP would weaken support and suggest alternative interpretation (e.g. O2/Ar-NCP is over-estimated).

**Analysis of NCP:**

I don't find the authors' analysis of NCP to be completely clear. First of all, I don't think that the net accumulation of POC should be called "NCP". I think that it would be clearer to call it "net POC accumulation" or "NCP minus export and DOC production" or something else. Second of all, I am not sure why the authors do not attempt to estimate NCP from nutrient drawdown. For example, the authors find a 0.9 $\mu$M drawdown of ML nitrate/nitrite over 3 days at site 1. If we naively convert that to carbon via redfield ratio

and ignore mixing, doesn't that imply NCP of 2$\mu$M C per day or 38 mmol/m2 over a 19 m ML? If so, why is this number so much lower than the mean daily O2/Ar-NCP of 150 mmol/m2 over this period? Am I missing something? Can nutrient supply from below plausibly explain the difference? Regardless, a second calculation of NCP, using NO3 drawdown, would be very helpful in interpreting the main results of this study, which all hinge on the accuracy of NCP via a single method (O2/Ar).

**Phytoplankton carbon diel cycles**:

To my knowledge there is no evidence that bbp diel cycles reflect diel cycles in phytoplankton biomass. In fact, I think that both previous literature and the results presented in this manuscript point to the opposite conclusion – bbp is not closely correlated with phytoplankton carbon on a sub-daily timescale. I therefore do not agree with the author's interpretation of diel cycles in bbp as diel cycles in phytoplankton biomass. My reasoning is as follows:

Phytoplankton carbon is produced during the day and consumed at night (through phytoplankton respiration and grazing), so we expect phytoplankton carbon to show a strong diel cycle, with a local maximum near dusk and a minimum near dawn. Repeated studies show that while cp diel cycles follow this expected pattern, bbp diel cycles often do not. During this study, diel cycles in bbp were observed in the upwelling site (similar to cp), but in the offshore site, diel cycles in cp were much stronger than bbp diel cycles. This is the opposite of what we would expect if bbp tracked phytoplankton carbon and cp tracked overall POC.

In addition to this empirical evidence, the latest theoretical work also does not suggest that bbp is more closely connected with phytoplankton carbon than cp. Coated sphere models suggest that roughly the same size classes of marine particles contribute to cp as to bbp (https://doi.org/10.1038/s41467-018-07814-6). If this is true, then differences in bbp and cp diel cycles should probably be interpreted as diel changes in particle composition (e.g. refractive index and/or internal structure) rather than a difference

between phytoplankton and POC diel cycles.

Graff et al. (2015) did find that phytoplankton carbon was better correlated with bbp (r2=0.69) than with cp (r2=0.42) across a range of mostly tropical and subtropical sites. However, these results say nothing about correlation on diel timescales and still imply that bbp is more strongly correlated with POC (r2=0.74) than with phytoplankton carbon. So while bbp may be a useful proxy of approximate phytoplankton carbon for the purpose of productivity modeling, this does not imply that bbp-based phytoplankton carbon (based on a single empirical study and not validated in this environment) is accurate enough track diel or even 2-3 day changes. So I would recommend removing or de-emphasizing the interpretation of bbp vis-à-vis phytoplankton carbon outside its more established use in the CbPM model.

**CDOM and DOC**:

The authors interpret differences in dissolved matter absorption as indicative of patterns in DOC (e.g. line 587). If these absorption measurements are correctly blanked (sometimes a challenge for AC-S data), then absorption of filtrate at 440 nm should indeed represent primarily colored dissolved organic matter (CDOM), which does contribute to DOC. However, it is my understanding that the fraction of DOM that absorbs light at 440 nm is relatively small, and that this fraction consists mostly of refractory, humic-like substances, whose dynamics are driven primarily by circulation and photodegradation, not necessarily by recent in-situ production. As such, the authors' claim that CDOM absorption measurements can be useful for tracking the partitioning of fixed carbon into DOC (line 766) is not really supported by the literature.

**Bbp spectral slope and particle size**:

The authors interpret changes in bbp spectral slope as indicative of changes in particle size. While this interpretation does have a clear theoretical basis, and while this interpretation is widely used in the literature, the authors should be cautioned that, again, there is no strong empirical support for this relationship. In fact, a recent extensive

study of bbp and particle size spanning a wide range of spectral slopes (-3 to 0) found no correlation with size at all.

**Line by line comments**:

Line 51: Claustre et al. 2007 (a Biogeosciences discuss paper) citation should be replaced with the 2008 peer-reviewed Biogeosciences citation.

Line 592: Do you mean "low overall DOC accumulation", rather than "low overall DOC concentration"? Most DOC is highly refractory with long residence, so we don't really know much about total DOC concentration from recent DOC production.
* * *

---

## Referee Comment (RC2) · Anonymous Referee #2 · 30 Aug 2019

This manuscript presents results from two Lagrangian 3-day experiments off the coast of Oregon conducted in two different environmental settings: coastal upwelling and off-shore conditions. The authors used the diel cycles of O2/Ar and beam attenuation (as a proxy for POC) to estimate mixed layer NCP, and attributed the large discrepancies between both methods to C export and DOC production. These relatively novel methods avoid the potential artifacts associated with incubations and their application from autonomous platforms (equipped with O2 sensors and transimissometers) have the potential to increase the spatial and temporal coverage of metabolic rates in the ocean, improving our understanding of the oceanic C cycle. Understanding the differences between O2-based and C-based diel rate estimates is therefore critical and as such I

think this manuscript will make an important contribution to the literature. However, I have a few general concerns that I think need to be addressed before publication.

General comments

Vertical mixing correction using N2O concentrations: given that the shallow mixed layer depths represented only a fraction of the euphotic zone, I wonder whether the N2O correction is suitable in this environment to estimate vertical mixing. Showing the N2O and O2 profiles (even if in the supplementary material) would be useful to assess this. As nitrification is photoinhibited, N2O concentrations typically start increasing below the euphotic zone. In addition, in the coastal experiment, I wonder if there could be other sources of N2O such as denitrification or lateral inputs.

POC vertical mixing correction: the correction for the vertical mixing of POC uses POC concentrations at the DCM to estimate the gradients, even though the mixed layer is much shallower. Please explain how this might affect the flux estimates.

In general, I think the authors should provide the air-sea and vertical flux terms used to estimate NCP so that the reader has an idea of the magnitude of the corrections (for example, the magnitude of these corrections could be included in Table 1).

To better understand the discrepancies between both methods, I recommend including in the introduction a more comprehensive description of the assumptions required for each method.

The diel approaches have been mostly used to estimate GPP and respiration (R). To estimate NCP it is probably more appropriate to use the real-time changes in O2/Ar or POC (as per Hamme et al., 2012). I recommend including NCP calculated this way. Also, I really encourage the authors to report GPP and R estimates based on the diel O2 and POC cycles, as these rates could provide some insight about the source of the discrepancies observed. For example, if we can assume that C export and DOC production are constant throughout the day, POC-GPP estimates would not be affected

by this carbon loss (see White et al., 2017, supplementary information) and therefore, POC-GPP should agree with O2/Ar-GPP. Under these circumstances POC-R would be overestimated and NCP underestimated. However, a diel cycle in C export, DOC production/consumption, and grazing would affect both GPP and R POC estimates. Again, a better description of how different processes might affect the diel cycle of POC would be useful.

The authors argue that NCP estimates from the different methods agree during the second experiment, even though they even show opposite directions and POC-NCP is <5% of O2/Ar-NCP. Are the uncertainties so high that the difference between both methods is within error? Again, I would compare GPP and R, as they will have lower uncertainty.

It is unclear to me why the drawdown of nutrients represents C export rather than NCP. Please elaborate on this. Also, do you expect the vertical mixing of nutrients to affect this estimate?

Given the wide range in the reported fraction of NCP that goes into the DOC pool, I doubt that using a value of 40% to estimate DOC production and C export is justified.

Specific comments

L47-49 I agree with reviewer 1 that the diel cycle is not needed to estimate NCP, but rather it is useful to estimate GPP and R

L50-51 In oligotrophic regions a significant fraction of phytoplankton total production goes to the DOC pool (Karl et al. 1998)

L60-61 " providing and indirect measure of carbon export out of the mixed layer". NCP is only equivalent to C export at steady state and over long timescales

L197-198 do you mean 0.125 kg/m3 instead of 0.25 kg/m3?

L299-300 was there a gradient in N2O below the mixed layer? If not, the lack of supersaturation might not be a good indicator of the absence of mixing.

L591-592 This sentence is unclear.

Table 1 Why are the NCP-POC results not included in the table? I suggest adding to this table all the terms used for the calculation of NCP, that is, the corrections for vertical mixing and air-water gas exchange

Figure 1. The resolution of this figure is not very good. The "x" symbol indicating the initial release of the drifter is hard to see.

---

## Author Comment (AC1) · 22 Oct 2019

Dear reviewer #1,

Thank you for your close reading of the manuscript, and for the depth of your feedback. Based on these comments, we have revised the manuscript to provide: (1) greater focus in the results and discussion sections, and clearer articulation of the study's contribution to the scientific community, (2) a more detailed quantification of the diurnal balance among the organic carbon (POC) and dissolved O2 sources and sinks (primary productivity, respiration, vertical mixing and gas exchange), and (3) a more conservative approach to interpreting estimates of organic carbon partitioning into DOC

production and particle export. In the following, the reviewer comments are shown in quotation marks followed by our response. In general, we have tried to respond to the comments in order; however, when several comments in one review are connected, we address them together. References cited in the responses are provided at the end of this document.

Reviewer #1 comments & responses:

". . . I think that the manuscript needs to be better focused on clearer conclusions that arise directly from the results of this study. Put another way, after reading the manuscript, I am not sure exactly in what way the authors think that the specific results of this study have advanced scientific knowledge. . . Also, please be specific about why the findings are important."

We believe that the main contributions of the work are as both a methodological "proof of concept" and as a source of new information on mixed layer carbon dynamics in the Subarctic Pacific. With the growing use of autonomous measurements of POC from optical beam attenuation (cp) and dissolved O2 on ships, moorings and gliders, there is greater potential to quantify ecosystem metabolism and carbon flows in the ocean mixed layer, with high spatial / temporal resolution, and without potential artifacts associated with incubations. Understanding the differences between O2-based and C-based estimates is critical in this regard. As these autonomous measurements have seldom been compared directly in the same time span and location, our study advances understanding of the ecological conditions in which they agree or diverge. In particular, our study directly expands the analyses reported by Alkire et al. (2012) and Briggs et al. (2018) from the North Atlantic Bloom to a new ocean environment, in which seasonal upwelling plays a key role in the productivity of several important fisheries. This upwelling environment presents an interesting methodological challenge because vertical mixing can affect cp and $\Delta$O2/Ar time series. Thus, our study further illustrates a new approach to applying mixing corrections to POC and O2 mass balances.

Based on the reviewer's comments, we have made an effort to emphasize the importance of these key results in the revised discussion and conclusion sections of the manuscript. Specifically, we have rewritten parts of the introduction to emphasize the goal of this study (to compare two autonomous measures of NCP in a new environment), and have rewritten parts of the discussion and conclusion to demonstrate how the results support this objective. We hope that these amendments clarify how our work advances scientific knowledge of ecosystem metabolism in the Northeast Pacific Ocean, and the methodologies used to quantify these metabolic rates.

"The interpretation of the various optical proxies should be tightened up, clarifying the extent of empirical support for each proxy, the use of the term "diel cycles" should be clarified. . ."

In the Sect. 4.2, we have added a paragraph discussing the uncertainty associated with several more optical proxies than discussed in the original manuscript, to ensure that readers understand both the strengths and limitations of our interpretations. As part of our response to reviewer #2 we have also made efforts to elaborate the assumptions and limitations of these proxies in the introduction.

The term "diel cycles" refers to a daily pattern of change. Because our measurements do not consistently display cyclical behavior, we realize that the term "diel cycles" may be misleading in this study. Therefore, we have rephrased the term diel cycles as "diurnal variations" in all instances. We feel that this term is appropriate for our continuous measurements made across several day – night transition periods.

". . .one of the conclusions seems to be that this method promises to expand the coverage of export estimates. How exactly (autonomous or ship-based), and what accuracy can we expect?"

We acknowledge that there are several limitations to accurately estimating POC export in our study, especially as a result of the uncertainties in vertical mixing correction and POC loss to the DOC pool. We have expanded uponÂăthese limitations in the
discussion and conclusion sections to remind readers of the caveats associated with our export estimates. As a result of these caveats, we removed POC export as quantity plotted in Fig. 4. In the conclusion we have further clarified how our work can inform future export estimates from autonomous or ship-based platforms.

"1. Is the main advance a "proof of concept" of the approach (simultaneously tracking net O2 production and net POC accumulation to constrain export) in two additional environments? If so, what are the criteria for success of this "proof of concept"? 2. Is the main advance some new knowledge about the functioning of the specific two ecosystems studied (i.e. Upwelling and offshore N. Pacific)? If so, what exactly have we learned and how does it differ from (or strengthen) previous understanding?... . . .the only conclusion in the conclusions section that clearly comes from the data presented in this manuscript is that O2 and POC cycling is more "coupled" offshore than in the upwelling region. But to me this statement is vague; I don't understand exactly what it means or why it's important. It would be much clearer to say something like "we find that a higher fraction of production is exported in region X than region Y" or "O2/Ar-based NCP can be used as a proxy for carbon export in region X but not region Y"."

The aim of our study is to show how simultaneous $\Delta$O2/Ar and POC time series measurements can provide quantitative estimates (with some caveats) of carbon export in marine regions of high vs. low productivity, and to suggest how such approaches may be valuable additions to more labor-intensive techniques such as the 238U- 234Th disequilibrium method. Specifically, we have endeavored to show that this approach is useful in an environment different from the North Atlantic bloom region. The major criterion for success of this "proof of concept" is the consistency of our measurements with what is known in upwelling and oligo/mesotrophic environments in the Northeast Pacific Ocean (for which there is a significant scientific literature). We expected more export in the upwelling site compared to the offshore waters, and therefore greater discrepancy between POC and $\Delta$O2/Ar diurnal variability, as observed in Alkire et al. (2012) and Briggs et al. (2018). We have made changes in the manuscript discussion and conclusions to make these criteria for proof of methodology more transparent, and have made our hypothesis regarding the differences in NCP comparisons in both drifter sites more explicit in the introduction. In particular, we have made efforts to explain why the contrasting results at the two drifter sites support the case for a proof of concept.

While we acknowledge that findings add little fundamental new knowledge to understanding of these two environments, this is not a primary conclusion of our study, as we have now clarified in the revised manuscript.

"For example, the authors tentatively conclude that their high O2/Ar-NCP estimates, if accurate, imply that both the widely-used CbPM model and the even more widely-used C-14 incubation method might be substantially under-estimating NPP in this environment. Even if this conclusion is not certain, if the authors believe that this is the most likely interpretation of their results, this conclusion is worth highlighting, because the accuracy of these NPP methods are of broad importance to the field."

We agree that this is an important result, and have emphasized it further in Sect. 4.3. But, because we do not think this is a main contribution of the study, given a limited data set to compare different NPP estimates, we did not add this result to the conclusions section.

"For NCP calculation, diel cycles (sub-daily data) are not really needed at all. All you need is the net change from the beginning (or end) of one day to the next. So the repeated claim that diel cycles are used to calculate NCP does not make much sense to me. Diel-cycle-based gross production estimates have their own uncertainties, but I think that they would be very valuable for the interpretation of the results of this study. The authors hypothesize that both NPP methods may be biased low. A finding that diel cycles-based gross production of O2 and POC are substantially higher than the NPP estimates would increase support for this hypothesis. A finding that gross production agrees with NPP would weaken support and suggest alternative interpretation (e.g. O2/Ar-NCP is over-estimated)."

NCP represents the difference between gross primary production (GPP) and commu-
nity respiration, which we have calculated from the best-fit slope from linear regression
of [O2]bio and POC against time within each day/night interval of both drifter deploy-
ments (Equations. 2, 8). Utilizing these sub-daily data maximizes the signal to noise
ratio in the time series, minimizing the error of the best-fit slope. The reviewer's sug-
gested approach is equivalent to taking the slope between two points at 24 hour in-
tervals, which would introduce greater uncertainty into the NCP calculation. We have
clarified this in the methods Sect. 2.6.

We have calculated two separate GPP approximations per drifter deployment day
(White et al. 2017; Claustre et al. 2008), and have added them to a revised Table
1. In Section 4.3, we compare GPP and net daily [O2]bio accumulation to NPP.

"I don't find the authors' analysis of NCP to be completely clear. First of all, I don't
think that the net accumulation of POC should be called "NCP". I think that it would be
clearer to call it "net POC accumulation" or "NCP minus export and DOC production"
or something else."

In regions where POC production and loss are in close balance, NCP has been derived
from diurnal variations in cp-derived POC (Claustre et al. 2008; White et al. 2017). For
consistency with these prior studies, we have chosen to use the term NCPPOC, but we
now indicate in the revised methods section 2.6 that this NCPPOC is more accurately
net POC accumulation.

"Second of all, I am not sure why the authors do not attempt to estimate NCP from nu-
trient drawdown. For example, the authors find a 0.9 _M drawdown of ML nitrate/nitrite
over 3 days at site 1. If we naively convert that to carbon via redfield ratio and ig-
nore mixing, doesn't that imply NCP of 2_M C per day or 38 mmol/m2 over a 19 m
ML? If so, why is this number so much lower than the mean daily O2/Ar-NCP of 150
mmol/m2 over this period? Am I missing something? Can nutrient supply from below
plausibly explain the difference? Regardless, a second calculation of NCP, using NO3

[Figure]

drawdown, would be very helpful in interpreting the main results of this study, which all hinge on the accuracy of NCP via a single method (O2/Ar)."

We thank the reviewer for this useful suggestion. As suggested, if we assume a Redfield ratio for organic matter production, cumulative nutrient (phosphate, silica, nitrate + nitrite) drawdown based on observations would equate to 36 to 50 mmol m-2 d-1 of organic carbon, which greatly underestimates NCPO2/Ar (Fig. 4). Most likely, upwelling of nutrient rich deep seawater has dampened the nutrient drawdown signal. Similar to our POC vertical mixing correction (Eq. 8), we have now applied the kmix term from Eq. 6 and calculated vertical gradient terms from nutrient profiles to correct for the effect of mixing on the apparent magnitude of nutrient drawdown. This calculation yields higher values, 101 to 132 mmol m-2 d-1 C, which are closer to NCPO2/Ar values, and thus provide support for our approach (Table 1). We address these calculations in the discussion section 4.1.1.

"So I would recommend removing or de-emphasizing the interpretation of bbp vis-à-vis phytoplankton carbon outside its more established use in the CbPM model."

We have removed all calculations of NCP based on diurnal variations in phytoplankton carbon, and have focused less on interpretations of diurnal variability in this backscatter-derived metric. We still report Cph concentrations in Fig. 2 to show the differences between drifter sites, but explain the limitations of interpreting such values in the discussion Sect. 4.2, as suggested by this reviewer.

"However, it is my understanding that the fraction of DOM that absorbs light at 440 nm is relatively small, and that this fraction consists mostly of refractory, humic-like substances, whose dynamics are driven primarily by circulation and photodegradation, not necessarily by recent in-situ production. As such, the authors' claim that CDOM absorption measurements can be useful for tracking the partitioning of fixed carbon into DOC (line 766) is not really supported by the literature."

Absorption at 440 nm by chromophoric dissolved organic matter has been observed to track seasonal variations in [Chl-a] in several locations, and a number of studies have suggested that absorption is impacted by microbial degradation of organic matter in the mixed layer, in addition to abiotic processed like photooxidation (e.g., Nelson et al. 1998; Organelli et al., 2014). However, we do agree that measurement of CDOM absorption at one wavelength (440 nm) is limiting, and that absorption at lower wavelengths (e.g., <400 nm) and the spectral slope of CDOM absorption over a range of lower wavelengths (e.g., 275 – 295 nm) would shed more light on the sources and sinks of CDOM in the mixed layer (Del Vecchio and Blough 2004; Grunert et al. 2018). We have thus amended the conclusion section to clarify that both CDOM absorption and spectral slope measurements would improve understanding in NCP transfer to the DOC pool on daily to seasonal time scales.

"The authors interpret changes in bbp spectral slope as indicative of changes in particle size. While this interpretation does have a clear theoretical basis, and while this interpretation is widely used in the literature, the authors should be cautioned that, again, there is no strong empirical support for this relationship. In fact, a recent extensive study of bbp and particle size spanning a wide range of spectral slopes (-3 to 0) found no correlation with size at all."

We acknowledge that there are many limitations to interpreting the particulate backscatter slope in terms of particle size distribution and have revised the manuscript to better address these limitations. However, we find that the link between slope and size is generally sensible in terms of differentiating both drifter sites. Moreover, the conclusions we derived about particle size distributions from the back-scatter slope were supported by results from size fractionated [Chl-a] measurements, and HPLC pigment-derived phytoplankton size distribution.

"Line 51: Claustre et al. 2007 (a Biogeosciences discuss paper) citation should be replaced with the 2008 peer-reviewed Biogeosciences citation."

We have changed the citation date to the 2008 Biogeosciences paper in all instances.

"Line 592: Do you mean "low overall DOC accumulation", rather than "low overall DOC concentration"? Most DOC is highly refractory with long residence, so we don't really know much about total DOC concentration from recent DOC production."

Yes, thanks for pointing this out. We have changed "concentration" to "accumulation".

References in author response

Alkire, M. B., D'Asaro, E., Lee, C., Jane Perry, M., Gray, A., Cetinić, I., Briggs, N., Rehm, E., Kallin, E., Kaiser, J. and González-Posada, A.: Estimates of net community production and export using high-resolution, Lagrangian measurements of O2, NO3−, and POC through the evolution of a spring diatom bloom in the North Atlantic, Deep Sea Res. Part I Oceanogr. Res. Pap., 64, 157–174, doi:10.1016/j.dsr.2012.01.012, 2012.

Briggs, N., GuÃřmundsson, K., Cetinić, I., D'Asaro, E., Rehm, E., Lee, C. and Perry, M. J.: A multi-method autonomous assessment of primary productivity and export efficiency in the springtime North Atlantic, Biogeosciences, 15(14), 4515–4532, 2018.

Claustre, H., Huot, Y., Obernosterer, I., Gentili, B., Tailliez, D. and Lewis, M.: Gross community production and metabolic balance in the South Pacific Gyre, using a non intrusive bio-optical method, Biogeosciences, 5, 463-474, 2008.

Del Vecchio, R., and Blough, N. V.: Spatial and seasonal distribution of chromophoric dissolved organic matter and dissolved organic carbon in the Middle Atlantic Bight, ÂăMar. Chem., Âă89(1-4), 169-187, 2004.

Grunert, B. K., Mouw, C. B., and Ciochetto, A. B.: Characterizing CDOM spectral variability across diverse regions and spectral ranges, ÂăGlobal Biogeochem. Cycles, Âă32(1), 57-77, 2018.

Nelson, N. B., Siegel, D. A. and Michaels, A. F.: Seasonal dynamics of colored dissolved material in the Sargasso Sea, ÂăDeep Sea Res. Part I Oceanogr. Res. Pap., Âă45(6), 931-957, 1998.

Organelli, E., Bricaud, A., Antoine, D. and Matsuoka, A.: Seasonal dynamics of light absorption by chromophoric dissolved organic matter (CDOM) in the NW Mediterranean Sea (BOUSSOLE site), Deep Sea Res. Part I Oceanogr. Res. Pap., 91, 72–85, 2014.

White, A. E., Barone, B., Letelier, R. M. and Karl, D. M.: Productivity diagnosed from the diel cycle of particulate carbon in the North Pacific Subtropical Gyre, Geophys. Res. Lett., 44(8), 3752–3760, 2017.

Please also note the supplement to this comment:
https://www.biogeosciences-discuss.net/bg-2019-257/bg-2019-257-AC1-supplement.pdf

———————————————

[Figure]

**Net community production**

Daily net community production (NCP)

**Fig. 1.** Daily net community production (NCP)

**Drifter 1**

**Drifter 2**

(a)

All measurements
Matched to drifter location (b)

(c)

**Fig. 2.** Time-series of ∆O2/Ar, POC concentration and Chl-a concentration

**Supplement:**

**Supplementary Information**

[Figure]

**Figure S1**: Vertical profiles of (left to right) dissolved nitrous oxide (N₂O) concentration, dissolved oxygen (O₂) concentration, [N₂O]bio (Sect. 2.6, Eq. 7), [O₂]bio (Sect. 2.6, Eq. 3), and the ratio of [O₂]bio to [N₂O]bio during (top) drifter period 1 and (bottom) drifter period 2. Gas samples were taken during CTD deployments from 12:00 to 14:00 of each day. The horizontal lines on each panel represent the average depths of the mixed layer (solid line) and base of the euphotic zone (dashed line). All concentration units are mmol m-3. The uncertainty in [O₂]bio/[N₂O]bio is

14.1% (Izett et al., 2018).

[Figure]

**Figure S2**: Relationship between particulate organic carbon (POC) concentration measured in 5

m CTD cast samples and underway beam attenuation ($c_{p660}$) at 660 nm, measured within 5

minutes of the cast time. The five higher values were collected during drifter period 1, while the two lowest values were collected during the first day of drifter period 2. The best-fit linear regression of [POC] against $c_{p660}$ (thick black line) is plotted relative to the linear regression (thin line) reported in Graff et al. (2015).

[Figure]

**Figure S3:** (a-c) Underway measurements of sea surface PAR, temperature, and salinity.
Anomalous values, likely from intrusion of external water masses into the sampled drifter patch,
are shaded. Underway observations from these periods were omitted from the data set. The brief
decrease in PAR during the morning of 21 August was caused by a total solar eclipse. (d) The
spectral slope of particulate backscatter ($b_{bp}$). (e) Bulk refractive index ($\eta_p$) at 470 nm, 532 nm
and 650 nm.

[Figure]

**Figure S4**: Nutrient concentration time series during both drifter periods. The gray point indicates an anomalously high value measured during an erratic CTD cast during the third night of drifter period 1. This data point was omitted from analysis. The dashed line is the best fit linear regression of each nutrient concentration against time. All regressions indicated significant changes (p<0.05), except for [NO3-+NO2-] concentrations during drifter survey 2.

[Figure]

**Figure S5**: Rates of change in (a) biological oxygen saturation ($\Delta O_2/Ar$), (b) particulate organic carbon (POC) concentration, (c) chlorophyll-a (Chl-a) concentration, and (d) the particulate backscatter ($b_{bp}$) slope. The rates were derived from linear regressions over successive day (D

open circle) and night (N shaded circle) intervals during the two drifter deployments. Slopes of significant linear regressions ($p<0.05$) are plotted as larger circles, and slopes of non-significant linear regressions ($p\geq0.05$) are plotted as smaller circles. The vertical bars span the lower and upper $95_{th}$ confidence intervals of the regression slopes.

---

## Author Comment (AC2) · 22 Oct 2019

Dear reviewer #2,

Thank you for your close inspection of this manuscript. Based on your comments, we have revised the manuscript to provide a more detailed quantification of the diurnal balance among the organic carbon (POC) and dissolved O2 sources and sinks (primary productivity, respiration, vertical mixing and gas exchange), and improved figures and table. In the following, the reviewer comments are shown in quotation marks followed by our response. In general, we have tried to respond to the comments in order; however, when several comments in one review are connected, we address them together.

References cited in the responses are provided at the end of this document.

Reviewer #2 comments & responses:

"Vertical mixing correction using N2O concentrations: given that the shallow mixed layer depths represented only a fraction of the euphotic zone, I wonder whether the N2O correction is suitable in this environment to estimate vertical mixing. Showing the N2O and O2 profiles (even if in the supplementary material) would be useful to assess this. As nitrification is photoinhibited, N2O concentrations typically start increasing below the euphotic zone. In addition, in the coastal experiment, I wonder if there could be other sources of N2O such as denitrification or lateral inputs."

As suggested, we have added N2O and O2 profiles to the supplement (Fig. S1), which indicate the depth of 1% of surface PAR as the base of the euphotic zone relative to the mixed layer depth. At drifter site 1, the average mixed layer depth (zmld = 19 m) is ~14 m shallower than the average euphotic zone depth (zeu = 33 m). Regardless, the new figure shows that mixing ratio [O2]bio/[N2O]bio (a key term of the mixing correction in Eqs. 5-6) remains relatively constant with depth below the mixed layer, suggesting that it is not significantly altered by biological processes between the zmld and zeu depths. We excluded two additional CTD casts during which N2O and O2 concentrations were measured on the reasoning that they were likely influenced by an external water mass. Denitrification is an unlikely additional source of N2O because measured water column O2 concentrations during CTD casts never fall below ~50 mmol kg-1 in the upper 100 m of the water column, well above the denitrification threshold (e.g., Hopkinson and Barbeau 2007). Notwithstanding the caveat that N2O profiles were only measured once per 24 hours, we neglect lateral advection of N2O into the drifter site because there is little change in [O2]bio/[N2O]bio between the two CTD casts used to apply a vertical mixing correction during drifter deployment 1, suggesting a water masses with similar N2O signatures. We discuss these assumptions in Sect. 2.6 of the revised manuscript.

"POC vertical mixing correction: the correction for the vertical mixing of POC uses POC concentrations at the DCM to estimate the gradients, even though the mixed layer is much shallower. Please explain how this might affect the flux estimates."

Our correction for vertical mixing of POC uses a gradient between the mixed layer depth and the shallowest depth at which transmissivity profiles, collected by the CTD, reach their maximum values (i.e., minimal particle concentrations), below which there is little change in apparent particle concentrations. We acknowledge that there is an error associated with this assumption, as beam transmissivity does not equate to POC concentration. Specifically, a different $\Delta POC/\Delta z$ gradient term affects the magnitude of the POC correctionĂă(Eq. 8), and thereby POC-derived NCP estimates. Although we do not account for this in our error propagation of NCP calculations, we have added an explicit assessment of this uncertainty in the methods and discussion.

"In general, I think the authors should provide the air-sea and vertical flux terms used to estimate NCP so that the reader has an idea of the magnitude of the corrections (for example, the magnitude of these corrections could be included in Table 1)."

We agree with the reviewer and have added these terms to Table 1.

"To better understand the discrepancies between both methods, I recommend including in the introduction a more comprehensive description of the assumptions required for each method."

As part of our response to reviewer #1 we have elaborated on the assumptions and limitations of the methodologies in the introduction. Several limitations were already explained in the discussion (Sect. 4.2).

"The diel approaches have been mostly used to estimate GPP and respiration (R). To estimate NCP it is probably more appropriate to use the real-time changes in O2/Ar or POC (as per Hamme et al., 2012). I recommend including NCP calculated this way. "

It is problematic in our dataset to calculate instantaneous NCP because of noisiness

in the time series. We calculated NCP at three-hour resolution, and have found values to generally approximate our daily NCP values (Table 1). We have stated this in the methods section 2.6. Hourly resolution was insufficient for removing noise because the average data measurement interval after quality control (Sect. 2.1) was ∼15 minutes. Thus, we maintain that there is value to calculating NCP over daily time scales using daily rates of change in $\Delta O2/Ar$ and [POC] (Claustre et al. 2008; White et al. 2017).

"Also, I really encourage the authors to report GPP and R estimates based on the diel O2 and POC cycles, as these rates could provide some insight about the source of the discrepancies observed."

As requested by reviewer #1, we have added GPP and R estimates to Table 1, and a discussion of how these separate estimates have contributed to the calculated NCP discrepancy in Sect. 4.1.

"Again, a better description of how different processes might affect the diel cycle of POC would be useful."

We include a brief sentence describing the diurnal variations that would affect POC concentrations in the discussion Sect. 4.1.

"The authors argue that NCP estimates from the different methods agree during the second experiment, even though they even show opposite directions and POC-NCP is <5% of O2/Ar-NCP. Are the uncertainties so high that the difference between both methods is within error? Again, I would compare GPP and R, as they will have lower uncertainty."

As Table 1 shows, the differences between NCP and GPP estimated by $\Delta O2/Ar$ and POC time series generally exceed the propagated uncertainties, while the differences between R estimated by both approaches were generally smaller than the uncertainties in R. We have added statements about NCP differences relative to error to discussion section 4.1.

"It is unclear to me why the drawdown of nutrients represents C export rather than NCP. Please elaborate on this. Also, do you expect the vertical mixing of nutrients to affect this estimate?"

We have corrected the manuscript, equating nutrient drawdown to NCP. As we wrote in our response to reviewer #1, vertical mixing dampens the magnitude of nutrient drawdown during drifter deployment 1. We have now added a vertical nutrient mixing correction to the results and discussion sections of the revised manuscript. The NCP estimates, derived from mixing-corrected nutrient drawdown, show good coherence with our ∆O2/Ar-based estimates, which is encouraging.

"Given the wide range in the reported fraction of NCP that goes into the DOC pool, I doubt that using a value of 40% to estimate DOC production and C export is justified."

We agree with that there is considerable uncertainty associated with the assumption that 40% of NCP is transferred to DOC, which affects our POC export calculations. We have added additional discussion to Sect. 4.1 addressing the implications of this uncertainty on POC export calculations.

"L47-49 I agree with reviewer 1 that the diel cycle is not needed to estimate NCP, but rather it is useful to estimate GPP and R."

As in our response to reviewer #1, we maintain that utilizing sub-daily underway ∆O2/Ar and cp time series to calculate NCP is advantageous over extracting time points every 24 hours to estimate NCP because the former approach reduces error in ∆O2 or ∆POC (Claustre et al. 2008; White et al. 2017). This is expressed in methods section 2.6, as well.

"L50-51 In oligotrophic regions a significant fraction of phytoplankton total production goes to the DOC pool (Karl et al. 1998)"

We amended this sentence as suggested by the reviewer.

"L60-61 " providing and indirect measure of carbon export out of the mixed layer". NCP

is only equivalent to C export at steady state and over long timescales"

We removed this clause from the sentence.

"L197-198 do you mean 0.125 kg/m3 instead of 0.25 kg/m3?"

No, we did mean 0.25. We found that a higher critical density difference matched with the visible inflection points in density profiles from CTD casts during drifter deployment 2. Smaller differences were insufficient to capture the depth of inflection. We have clarified this in the methods section 2.3.

"L299-300 was there a gradient in N2O below the mixed layer? If not, the lack of super-saturation might not be a good indicator of the absence of mixing."

There is a vertical gradient in N2O from surface to below the mixed layer (Fig. S1 in revised manuscript). According to Izett et al. (2018), the sub-saturation of [N2O]bio in the mixed layer at drifter site 2 invalidates the vertical mixing approach in this environment. Furthermore, satellite-derived and underway temperature imply little upwelling at drifter site 2 (Figs. 1, S2). This observation is consistent with expectations of more stratified offshore water columns with limited vertical mixing.

"L591-592 This sentence is unclear."

Reviewer #1 had similar thoughts, and in response we changed the word "concentration" to "accumulation", attempting to clarify that a higher DOC/NCP ratio would still result in little absolute DOC accumulation rates at drifter site 2.

"Table 1 Why are the NCP-POC results not included in the table? I suggest adding to this table all the terms used for the calculation of NCP, that is, the corrections for vertical mixing and air-water gas exchange"

We have added all these components to Table 1.

"Figure 1. The resolution of this figure is not very good. The "x" symbol indicating the initial release of the drifter is hard to see."

We have fixed a typo in the description for this figure. The diamond symbol shows the location of initial drifter release in the center panel; there is no "x" symbol. We have made the diamond red and increased the font size of the color bar to make this figure clearer.

References in author response

Claustre, H., Huot, Y., Obernosterer, I., Gentili, B., Tailliez, D. and Lewis, M.: Gross community production and metabolic balance in the South Pacific Gyre, using a non intrusive bio-optical method, Biogeosciences, 5, 463-474, 2008.

Hopkinson, B. M., and Barbeau, K.A.: Organic and redox speciation of iron in the eastern tropical North Pacific suboxic zone, Âă Mar. Chem., Âă 106(1-2), 2-17, 2007.

Izett, R. W., Manning, C. C., Hamme, R. C. and Tortell, P. D.: Refined estimates of net community production in the Subarctic Northeast Pacific derived from $\Delta O2/Ar$ measurements with N2O‐based corrections for vertical mixing, Global Biogeochem. Cycles, 32(3), 326–350, 2018.

White, A. E., Barone, B., Letelier, R. M. and Karl, D. M.: Productivity diagnosed from the diel cycle of particulate carbon in the North Pacific Subtropical Gyre, Geophys. Res. Lett., 44(8), 3752–3760, 2017.

Please also note the supplement to this comment:
https://www.biogeosciences-discuss.net/bg-2019-257/bg-2019-257-AC2-supplement.pdf
* * *
[Figure]

**Sea surface temperature (°C)**

12    14    16    18

(a)

Latitude (°N)

oNewport

Longitude (°E)

(b) Drifter Period 1

(c) Drifter Period 2

**Fig. 1.** Drifter deployment map

[Figure]

**Fig. 2.** Time-series of ∆O2/Ar, POC concentration and Chl-a concentration

**Net community production**

$$mmol\ C\ m^{-2}\ d^{-1}$$

Legend:
- NCP$_{O2/Ar}$ (black)
- NCP$_{POC}$ (gray)

x-axis labels: Aug20 20:00, Aug21 20:00, Aug22 20:00, Aug24 7:30, Aug25 6:30

y-axis values: 200, 150, 100, 50, 0, -50, -100

**Fig. 3.** Net community production (NCP)

---

## Author Response (AR3)

Dear Dr. Marañón,

We thank the reviewer for their diligence in revising and strengthening our manuscript. Following their suggestions, we have modified the text by decreasing the emphasis on calculating export fluxes, reorganizing our discussion of the discrepancies between different productivity estimates and clarifying our conclusions. We also addressed the uncertainty in our calculations derived from the choice of respiratory quotients in the conversion of $\Delta O_2/Ar$-based calculations to molar carbon units. Finally, we have corrected typos throughout the entire manuscript. In the following, the reviewer's comments are shown in blue italicized text, with our responses below in non-italicized black text. References cited in the comments are listed in the manuscript. Please note that line numbers referenced below correspond to those in the revised manuscript that has been uploaded as a separate file. An additional version of the revised manuscript has been appended after our reviewer responses, with major revisions shown using the tracked changes function in MS Word.

*However, I do think that the discussion and conclusions could be better organized and more concise. I find the discussion hard to follow at times, and I think the authors could narrow down a bit better what the most plausible processes causing the observed discrepancies are (and this should be reflected in the conclusions).*

*As I view it, in drifter 1, the similarities in CR but discrepancies in GPP indicate that C export alone cannot explain GPP discrepancies. Granted that there could be enhanced export during daytime, but given how similar CR rates are it would mean that the export is pretty much halted at night (actually sometimes CR-O2/Ar is larger than CR-POC). A more plausible explanation for these observations is a combination of C export and DOC dynamics. The authors mention DOC production but mostly just as a "POC loss" term, whereas the POC method is really missing a combination of DOC production and DOC respiration. If there is net DOC production during the day from newly fixed C (see Karl et al., 1998) you would expect GPP-POC to be underestimated, and respiration of fresh DOC at night (missed by the POC method) would result in an underestimation of CR-POC. If C export (that would result in an overestimation of CR-POC) is of similar magnitude than DOC respiration, CR-POC would end up being of similar magnitude than CR-O2, but differences in GPP would be greater due to DOC production. The*

*alternative explanation would be the proliferation of large diatoms not captured by the beam*

*attenuation measurement, but the decrease in chla does not really support this idea.*

We have condensed and reorganized our main interpretations throughout the discussion. Sect. 4.1 is now split into two rather than three sub-sections according to drifter period, and as the reviewer recommended, we have focused on the major mechanisms that could have led to discrepancies in GPP, CR and NCP over day and night intervals. Discussion of these mechanisms are described first in more detail for the drifter 1 data (Sect. 4.1.1), and again, in less detail, for drifter 2 data (Sect. 4.1.2). The decoupling mechanisms are also highlighted in our revised conclusions (lines 923-926). We removed Sect. 4.4, relocating parts of the text in other sections of the discussion and conclusions. In consolidating the discussion, we further deleted a paragraph about error in the POC mixing correction from Sect. 4.2, and instead referenced its main point in Sect. 2.7 (lines 457-460). Overall, the length of the discussion has been reduced by 36% (from 5,417 words to 3,457 words).

*I do not totally understand how the correlation of 3h-NCP from both methods (Figure 5b) indicates that there are sub-daily variations in POC losses, even though I agree that potential daily changes in export could contribute to the differences. A graph showing the difference between 3-hour DeltaO2/Ar and DeltaPOC as a function of time of the day might be more useful to prove this.*

We have added the plot suggested by the reviewer to Fig. 5 and referenced it in both Sect. 4.1.1 and 4.11.2 (lines 691, 753, respectively).

*In drifter 2, I think that DOC dynamics alone could very well explain the differences observed. I found the discussion of the daily changes in heterotrophic biomass as a potential cause for the observed differences interesting, and I wonder whether this needs to be brought up before, with drifter 1, as it could also be affecting the differences in that case, combined with C export and DOC dynamics.*

We have now moved our discussion of the effects of variable heterotrophic biomass earlier into

Sect. 4.1.1 (lines 729-736) in the context of drifter site 1. We also discuss this in the context of drifter site 2, along with DOC production and carbon export (lines 774-789).

*Because the authors did not measure net DOC accumulation, and the range in the reported*

*fraction of NCP that goes into the DOC pool is quite wide, I do not think it is justifiable to*

*estimate POC export based a chosen DOC/NCP value.*

We have shortened our discussion and calculation of export fluxes, and removed all estimates of export from Sect. 4.1. Nonetheless, because we affirm the future potential for estimating export with these coupled methods, we still provide a condensed version of our export estimates just for drifter site 1 in the conclusions (Sect. 5, lines 942-945).

*L479 Is a PQ of 1.1 is justifiable for NCP? If we assume that NCP approximates new production*

*it is probably closer to 1.4.*

We have assumed that drifter site 2 represents an environment with a tight microbial loop and relatively stable mixed layer, with low $NO_3$ concentrations and much of the photosynthetic production fueled by $NH_4$. Under these conditions, we feel that a PQ value of 1.1. is, indeed, appropriate (Laws 1991). We have not changed this in the revised manuscript, as it is already clarified in Sect. 2.6 (lines 311-313).

*Equations: It would be helpful to specify the units of each term as well as the direction of the*

*fluxes. I understand that tD and tN sum 1 (day), in which case (tD + tN) does not need to be*

*added to the denominator in equations 2c and 9c. Or for consistency should be added to the*

*denominator in equations 2a and 9a.*

The terms $t_D$ and $t_N$ correspond to fractions of one day. On line 293, we have clarified that the *dt*

term in the Sect. 2.6 equations has units of days (and thus rates of change are per day). We agree with the reviewer and we have added $(t_D + t_N)$ to Eqs. 2 and 9 to convey that the productivity estimates are extrapolated over one full day. Additionally, we have clarified the direction of mixing and gas exchange in lines 309-311 of the revised manuscript, and units for terms when
relevant (lines 311, 345).
*Eq1: shouldn't Fmix be subtracted? (positive dO2/dz means an oxygen flux into the mixed layer).*
The depth gradient is calculated as deep minus surface values, and thus yields a negative term.
By subtracting this negative term in equations 1 and 8, we are effectively adding it to the
calculated NCP value. This has been fixed in the revised manuscript equations and in Table 1.
*Eq 2b why is the PQ used to convert CR? A RQ should be used instead.*
The respiratory quotients reported in past studies encompass a wider range than photosynthetic
quotients (Laws 1991), from values as low as ~0.5 to as high as ~1.7 (Anderson and Sarmiento
1994; Robinson et al. 1999; Lønborg et al. 2011; Daneri et al. 2012; Fernández-Urruzola et al.
2014). Because 1.1-1.4 is a common RQ range reported in literature (Anderson and Sarmiento
1994; Robinson and Williams 1999; Robinson et al. 2002; Hedges et al. 2002), we have opted to
assume that the RQ value approximates our chosen PQ for each drifter site. We have clarified
this in Sect. 2.6 of the revised manuscript (lines 313-322), and describe in greater detail how this
assumption may impact the results in Sect. 4.2 (lines 794-807).
Furthermore, it is not possible to apply an RQ value to several oxygen productivity terms in the
manuscript, especially the mixing term, which was not differentiated between day and night. The
fact that the ratio of the $\Delta O_2/Ar$ mixing term to the POC-based mixing term is roughly equivalent
to the selected PQ at drifter station 1 (mean $\pm$ 1 S.D. = 1.5 $\pm$ 0.2) affirms that this conversion
factor between mixing terms is reasonable.
*Eq 3: The percent symbol (%) is not needed. ΔO2/Ar should be defined somewhere*
In Sect. 2.2, we have defined $\Delta O2/Ar = 100\% * ([O2/Ar]_{meas} / [O_2/Ar]_{eq} - 1)$ (Tortell 2005;
Tortell et al. 2011). For this reason, we do not repeat the equation in Sect. 2.6.1 and the 1/100%
term is required in Eq. 3.

*Eq 9a, 9b: why are these divided by the PQ? they are already in C units.*

We have removed this PQ term. We thank the reviewer for catching this typo.

*L780 This is a weird sentence for the results section, as the conclusions have not been presented*
*yet.*

We have removed the term "conclusions" from the sentence and rephrased it in line 628 of the
revised manuscript.

*L931-936 I find this paragraph confusing. Without looking at the discrepancies or similarity of*
*GPP and CR you would be unable to know what is causing the discrepancies in NCP.*

In this paragraph, we mean to convey that assumption of a constant daily respiration rate can
lead to erroneous interpretations of GPP and CR. In the revised manuscript, we have moved this
point to Sect. 5 as a concluding remark (lines 926-928).

*Table 1. I would delete POC export estimates as the authors did not measure DOC production.*
*Also the numbers do not seem to match my calculations.*

These estimates have been deleted from the table.

*Table 2. Why is the Export+DOC column only added to drifter 1?*

We have filled in a similar column for drifter 2 to highlight the smaller discrepancy between
NCP measures during this drifter deployment.

[revised manuscript text omitted]